# A daily diary study on adolescents' mood, empathy, and prosocial behavior during the COVID-19 pandemic

Suzanne van de Groep[1,2]*, Kiki Zanolie[2,3], Kayla H. Green[1,2], Sophie W. Sweijen[1,2], Eveline A. Crone[1,2]

1 Erasmus School of Social and Behavioral Sciences, Erasmus University Rotterdam, Rotterdam, The Netherlands, 2 Leiden Institute for Brain and Cognition, Leiden University, Leiden, The Netherlands, 3 Department of Developmental and Educational Psychology, Institute of Psychology, Leiden University, Leiden, The Netherlands

* s.vandegroep@essb.eur.nl

**Data Availability Statement:** The data are available at DOI 10.25397/eur.12783161.

## Abstract

Adolescence is a formative phase for social development. The COVID-19 pandemic and associated regulations have led to many changes in adolescents' lives, including limited opportunities for social interactions. The current exploratory study investigated the effect of the first weeks of COVID-19 pandemic lockdown on Dutch adolescents' ($N$ = 53 with attrition, $N$ = 36 without attrition) mood, empathy, and prosocial behavior. Longitudinal analyses comparing pre-pandemic measures to a three-week peri-pandemic daily diary study showed (i) decreases in empathic concern, opportunities for prosocial actions, and tension, (ii) stable levels of social value orientation, altruism, and dire prosociality, and (iii) increased levels of perspective-taking and vigor during the first weeks of lockdown. Second, this study investigated peri-pandemic effects of familiarity, need, and deservedness on giving behavior. To this end, we utilized novel hypothetical Dictator Games with ecologically valid targets associated with the COVID-19 pandemic. Adolescents showed higher levels of giving to a friend (a familiar other, about 51% of the total share), a doctor in a hospital (deserving target, 78%), and individuals with COVID-19 or a poor immune system (targets in need, 69 and 63%, respectively) compared to an unfamiliar peer (39%) This suggests that during the pandemic need and deservedness had a greater influence on adolescent giving than familiarity. Overall, this study demonstrates detrimental effects of the first weeks of lockdown on adolescents' empathic concern and opportunities for prosocial actions, which are important predictors of healthy socio-emotional development. However, adolescents also showed marked resilience and a willingness to benefit others as a result of the lockdown, as evidenced by improved perspective-taking and mood, and high sensitivity to need and deservedness in giving to others.

**Funding:** This work was supported by an innovative ideas grant of the European Research Council (ERC CoG PROSOCIAL 681632 to E.A.C.)

**Competing interests:** The authors have declared that no competing interests exist.

# Introduction

Adolescence, the age period of approximately 10–24 years, is a transition phase for socio-emotional development in which adolescents learn to autonomously navigate their social worlds [1]. Adolescence is marked by heightened emotional reactivity and a reorientation to peers [1–3] such that the social world and interactions with people outside the household become increasingly important [1, 2]. Not only do adolescents start spending more time with peers compared to children, they also are more sensitive to peer influence and attach more value to peer approval, acceptance, and rejection [2–4]. Simultaneously, cognitive processes, such as perspective-taking and thinking about oneself in relation to others, improve. Perspective-taking enables adolescents to show other-oriented, prosocial behaviors (i.e., behaviors that benefit others) and to build strong, reciprocal relationships with others [1, 2, 4–6]. Improvements in social perspective-taking help adolescents grow into contributing members of society that have secure relationships with others [7, 8]. As such, opportunities to empathize, socialize, and benefit others are paramount for healthy adolescent development [1, 2, 8].

## Pre- to peri-pandemic changes in mood, empathy, and prosocial behavior

The COVID-19 pandemic in 2020 represents a massive and challenging global health crisis [9]. Without a vaccine, most governments have authorized containment measures, such as social distancing, to slow the pandemic [9]. However, distancing conflicts with the human tendency to connect with others and severely limits opportunities to have contact with people outside one's household. The need to connect with others is especially pronounced in adolescence [1, 2, 9]. In the Netherlands, social distancing and a lockdown were implemented on March 15, 2020, and schools were closed on March 17, 2020. This led to an enormous change in adolescents' social environment, with them fully transitioning to online home-education, having no physical contact with peers and friends, and being constrained to stay mostly at home. Concerns have been raised about how feelings of social isolation and loneliness will affect adolescents' mental health and socio-emotional development [2, 9]. For example, animal research has shown detrimental effects of social deprivation and isolation [2, 10, 11]. Besides social distancing, the pandemic has resulted in many other changes for adolescents, including possible worry about health of family members, fear of death, financial consequences, and worry about one's future [9, 12]. Therefore, social changes in the pandemic should be interpreted in the context of multiple system disruptions [13]. To date, very little is known about how the COVID-19 pandemic affects adolescents and their social behavior. Therefore, the first part of the current study was dedicated to investigating changes in Dutch adolescents' mood, empathy, and prosocial behaviors when comparing pre-pandemic measures to measures obtained in the first weeks of lockdown.

## Peri-pandemic giving to others

Adolescence is a crucial turning point for the development of other- and society-oriented, prosocial behaviors [6, 8]. Showing prosocial behaviors fulfills adolescents' need for autonomy, respect, and impact [14], and helps them to attain the fundamental developmental task of fitting in and being a contributing member of society [7, 8]. Giving is a form of prosocial behavior that is important for building and maintaining strong relationships [5, 15]. Dictator Games, in which individuals divide valuable resources with a second person [16], are often used to study giving, showing that individuals typically give away 20–30% of their resources [5, 17], even when the identity of the receiver (i.e., target) remains unknown and without future transactions. So far, most studies have focused on giving to unfamiliar others, but adolescents generally give more to close, relational others, such as friends and classmates [5, 15, 18].

Additionally, adults studies show increased levels of giving to those who are in need or deserving [7, 19, 20], although familiarity, need, and deservedness are no prerequisites for giving [7]. Relational giving and giving to those who are in need or deserving are nonetheless important to study, as they may be more prevalent in adolescents' lives than giving to unfamiliar, anonymous others [5]. The COVID-19 pandemic represents a particularly interesting and unique context to extend our knowledge about adolescents' giving to various targets, which so far has mainly focused on peers [5, 14, 17]. Specifically, COVID-19 introduces ecologically valid targets that are in need or deserving. These targets of interest include individuals with a poor immune system or COVID-19 symptoms (i.e., targets in need) and medical personnel, such as doctors working in hospitals (i.e., deserving targets). We compared giving to these three targets with giving to an unfamiliar peer (i.e., the default option in most economic games) and a friend (i.e., a target that is more familiar and similar). The reason to examine giving towards targets with verifying degrees of familiarity, similarity, and need and deservedness within individuals was to get more insight in the mechanisms and motivations underlying giving decisions. Consequently, the second part of the current study was dedicated to investigating adolescents' peri-pandemic giving towards the following targets in hypothetical Dictator Games: an unfamiliar peer, friend, individual with a poor immune system, individual with COVID-19, and doctor working at a hospital. We also examined whether giving changed as a result being in lockdown for several weeks.

## The current study

In the current exploratory study, we investigated the effect of the COVID-19 pandemic and the associated lockdown and social distancing regulations on Dutch adolescents' mood, empathy, and prosocial behavior. Secondly, we investigated giving towards targets that play an important role during the COVID-19 pandemic in terms of familiarity, need, and deservedness, and whether this changed during the lockdown. We focused on adolescents aged 10–20 years, given that this is a crucial developmental phase for social behavior. Adolescents who already participated in a cohort-sequential longitudinal study prior to the pandemic were invited to participate in the current daily diary study. Participants reported on their experiences and behaviors on weekdays during the lockdown, from March 30 until April 17, 2020. Adolescents reported on their levels of empathy (i.e., perspective-taking and empathic concern), mood (i.e., vigor and tension) and prosocial behaviors (i.e., general contributions to society, opportunities for prosocial actions, social value orientation, altruism, and dire prosociality) prior to and peri-pandemic. Furthermore, at the start and end of the daily diary study, participants played single-shot Dictator Games in which they divided 10 coins between themselves and the following targets: an unfamiliar peer, friend, individual with poor immune system, individual with COVID-19, and a doctor working in a hospital.

## Method

### Participants

The current study was part of a larger longitudinal study on prosocial development in adolescence called 'Brainlinks'. In this cohort-sequential longitudinal study conducted in the Netherlands, adolescents are being followed over the course of several years to investigate the neural development of prosocial behavior, see Fig 1. At the first measurement wave (i.e., T1, May–October 2018) 142 adolescents participated, 127 of whom also participated in the second wave (i.e., T2, August 2019 –January 2020). In between these waves, all participants were asked to report on their mood and prosocial experiences every two months, (i.e., T1.5). At the start of the COVID-19 pandemic in 2020, all 142 original participants were contacted to participate in

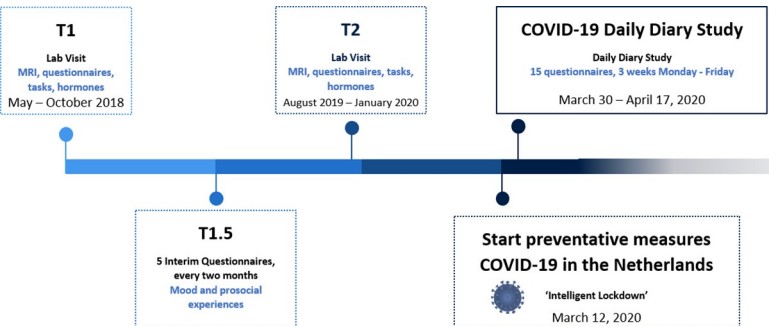

**Fig 1. Overview of the longitudinal Brainlinks study, aimed at investigating the predictors and consequences of being prosocial in adolescence.** The current study focuses on comparing pre- (i.e., T1, T1.5, and T2) to peri-COVID-19-pandemic behavior, as well as understanding adolescents peri-pandemic giving behavior.

a daily diary study which focused on adolescents' peri-pandemic prosocial experiences and behavior. A total of 53 adolescents participated, 42 (79.2%) of whom were female, and 11 (20.8%) were male. Participants were between 10–20 years of age on the first day of the daily diary study (i.e., March 30, 2020), $M_{age} = 16.56$, $SD = 2.67$, see S1 Fig. This study was approved by the METC Leiden-Den Haag-Delft (Leiden); number NL62878.058.17 and the Psychology Research Ethics committee of Universiteit Leiden, Name and Number: 'Prosocial behavior in adolescence during the pandemic Covid-19 crisis' (2020-03-25-Crone, prof.dr. E.A.M.-V2-2317). Adolescents and their parents (for participants younger than 16 years) provided written informed consent. Participants received €1 for each completed questionnaire and €2 for the last questionnaire. Additionally, to increase the response rate for the final questionnaire, a €50 gift card was allotted amongst the participants who completed this questionnaire.

## Procedure

**Daily diary study conducted during the COVID-19 pandemic.** Data were collected through online questionnaires in Qualtrics. Data collection started on March 30, 2020 (i.e., two weeks after the schools were closed in the Netherlands), and was completed on April 17, 2020. During these three consecutive weeks participants received an invitation by e-mail from Mondays to Fridays, to complete a set of questionnaires. Each questionnaire included daily measures (e.g. assessing mood and daily prosocial experiences during the COVID-19 pandemic; duration five minutes). Furthermore, on day 1, 5, 10, and 15, weekly measures were also collected (e.g. empathy, general contributions to society; duration five minutes). Finally, a few extra measures were collected only at the start (day 1) and end of the study (day 15; i.e., assessing giving behavior, social value orientation, and risk-taking behavior; duration five minutes). Therefore, the total duration of the set of questionnaires was 15 minutes on day 1 and 15, 10 minutes on day 5 and 10, and 5 minutes on all other days.

All questionnaires were sent out at 10:00 AM and participants were asked to complete the questionnaire that same day. Participants were instructed not to discuss the questionnaires with others and to always select the answer that described them best. The following inclusion criteria were used for the daily measures: (1) questionnaires had to be completed within 36 hours after being sent, (2), a response should not be a duplicate (i.e., a double response on the same questionnaire by one individual). In the case of duplicate responses where one was incomplete, the incomplete response was removed. In case of duplicates where the second response was obtained one day after the other, this response was moved to the correct day. The number of valid responses for all obtained measures can be seen in Table 1. As this number

**Table 1. Descriptive statistics of giving, empathy (i.e., perspective-taking and empathic concern), general contributions to society, opportunities for prosocial actions,, mood (i.e., vigor and tension), altruistic and dire prosociality, and social value orientation prior to and during the COVID-19 pandemic.**

| Measure | No. of items | Min. Score | Max. Score | Mean Score (SD) | Skewness | Kurtosis |
|---|---|---|---|---|---|---|
| *Donated Coins Daily Diaries Day 1 (N = 53)* | | | | | | |
| Unfamiliar Peer | 1 | 0 | 6 | 3.91 (1.62) | -1.42 | 1.00 |
| Friend | 1 | 3 | 10 | 5.11 (.89) | 3.16 | 17.74 |
| Individual with poor immune system | 1 | 4 | 10 | 6.25 (1.54) | -.25 | -.54 |
| Individual with COVID-19 | 1 | 3 | 10 | 6.89 (1.86) | .51 | -.61 |
| Doctor working at a hospital | 1 | 3 | 10 | 7.79 (1.78) | 1.04 | .67 |
| *Donated Coins Daily Diaries Day 15 (N = 36)* | | | | | | |
| Unfamiliar Peer | 1 | 0 | 6 | 4.25 (1.27) | -1.56 | 2.29 |
| Friend | 1 | 4 | 10 | 5.31 (1.06) | 3.10 | 11.36 |
| Individual with poor immune system | 1 | 3 | 10 | 5.67 (1.53) | .17 | -1.18 |
| Individual with COVID-19 | 1 | 1 | 10 | 6.25 (1.95) | -.23 | 1.02 |
| Doctor working at a hospital | 1 | 4 | 10 | 7.44 (1.84) | .55 | .64 |
| *Empathy: Perspective-taking* | | | | | | |
| T1 (N = 51) | 6 | 1.17 | 3.50 | 2.51 (.60) | -.45 | -.66 |
| T2 (N = 53) | 6 | 1.00 | 3.83 | 2.58 (.58) | -.25 | .95 |
| Daily diaries (average day 1, 5, 10, 15; N = 53) | 6 | 1.04 | 3.94 | 2.75 (.62) | -.47 | .33 |
| *Empathy: Empathic Concern* | | | | | | |
| T1 (N = 51) | 6 | 1.33 | 3.83 | 2.68 (.59) | .07 | -.62 |
| T2 (N = 53) | 6 | 1.00 | 4.00 | 2.63 (.66) | -.00 | -.13 |
| Daily diaries (average day 1, 5, 10, 15; N = 53) | 6 | 1.13 | 3.04 | 2.32 (.42) | -.58 | -.04 |
| *General Contributions to Society T2 (N = 53)* | | | | | | |
| T2 (N = 53) | 3 | 3.67 | 10.00 | 7.18 (1.16) | -.40 | .78 |
| Daily diaries (average day 1, 5, 10, 15; N = 53) | 3 | 3.50 | 9.42 | 7.25 (1.14) | -.65 | 1.09 |
| *Opportunities for prosocial actions* | | | | | | |
| T1 (N = 51) | 3 | 1.00 | 5.67 | 3.55 (1.43) | -.33 | -.10 |
| T1.5 (N = 51) | 3 | 1.17 | 5.92 | 3.75 (1.05) | -.21 | -.22 |
| T2 (N = 53) | 3 | 1.00 | 6.00 | 3.94 (1.18) | -.53 | -.04 |
| Week 1 (N = 52) | 3 | 1.00 | 5.00 | 3.03 (1.20) | -.08 | -1.04 |
| Week 2 (N = 39) | 3 | 1.00 | 5.33 | 2.80 (1.32) | .27 | -.99 |
| Week 3 (N = 40) | 3 | 1.00 | 5.80 | 2.96 (1.22) | .10 | -.36 |
| *Mood: Vigor* | | | | | | |
| T1.5 (N = 43) | 5 | 1.58 | 3.67 | 2.75 (.49) | -.43 | .22 |
| Week 1 (N = 52) | 5 | 1.44 | 4.60 | 3.12 (.81) | -.18 | -.56 |
| Week 2 (N = 39) | 5 | 1.64 | 5.00 | 3.33 (.81) | -.09 | -.10 |
| Week 3 (N = 40) | 5 | 1.40 | 5.00 | 3.25 (.90) | -.13 | -.16 |
| *Mood: Tension* | | | | | | |
| T1.5 (N = 43) | 6 | 1.00 | 4.33 | 2.15 (.85) | .58 | -.52 |
| Week 1 (N = 52) | 6 | 1.00 | 4.11 | 1.77 (.76) | 1.15 | .80 |
| Week 2 (N = 39) | 6 | 1.00 | 2.67 | 1.45 (.46) | 1.20 | .80 |
| Week 3 (N = 40) | 6 | 1.00 | 3.44 | 1.65 (.67) | 1.04 | .36 |
| *Dire prosociality* | | | | | | |
| T1 (N = 52) | 3 | 2.67 | 5.00 | 3.84 (.70) | .047 | -.79 |
| T2 (N = 51) | 3 | 2.00 | 5.00 | 3.77 (.83) | -.46 | -.43 |
| Week 1 (N = 52) | 3 | 1.56 | 5.00 | 3.81 (.81) | -.49 | -.13 |
| Week 2 (N = 39) | 3 | 1.67 | 5.00 | 3.70 (.90) | -.36 | -.51 |
| Week 3 (N = 40) | 3 | 1.42 | 5.00 | 3.67 (.89) | -.68 | .40 |

(*Continued*)

**Table 1.** (Continued)

| Measure | No. of items | Min. Score | Max. Score | Mean Score (SD) | Skewness | Kurtosis |
|---|---|---|---|---|---|---|
| *Altruistic prosociality* | | | | | | |
| T1 (N = 52) | 6 | 1.83 | 5.00 | 4.28 (.59) | -1.84 | 4.98 |
| T2 (N = 51) | 6 | 2.00 | 5.00 | 4.25 (.52) | -1.54 | 5.45 |
| Week 1 (N = 52) | 6 | 2.50 | 5.00 | 4.11 (.65) | -.79 | .04 |
| Week 2 (N = 39) | 6 | 2.63 | 5.00 | 4.17 (.63) | -.70 | .00 |
| Week 3 (N = 40) | 6 | 2.67 | 5.00 | 4.19 (.66) | -.89 | -.08 |
| *Social Value Orientation (SVO)* | | | | | | |
| T1 | 6 | 14.74 | 45.00 | 34.05 (7.30) | -.80 | .52 |
| T2 | 6 | 7.82 | 61.39 | 35.50 (7.91) | -.09 | 3.91 |
| Day 1 daily diary study | 6 | 7.82 | 53.37 | 35.12 (7.43) | -.68 | 2.94 |
| Day 15 daily diary study | 6 | 16.14 | 45.89 | 35.79 (6.80) | -.93 | 1.43 |

Note: Minimum and maximum scores represent the actual (i.e., not possible) scores that were made by participants.

differs between time points and measures, the number of participants will be reported for all analyses.

**Questionnaire data obtained before the COVID-19 pandemic.** At T1 (i.e., May–October 2018), T1.5 (between T1 and T2), and T2 (August 2019—January 2020), prior to the COVID-19 pandemic, we measured empathy (i.e., perspective-taking and emphatic concern), general contributions to society, opportunities for prosocial actions, dire and altruistic prosociality, SVO, and mood (i.e., vigor and tension), see Fig 1 and Table 1. Specifically, empathy, altruistic and dire prosociality, and SVO were measured at T1 (N = 51) and T2 (N = 53), general contributions to society at T2, (N = 52), opportunities for prosocial actions at T1 (N = 51), T1.5 (N = 51), and T2 (N = 53), and mood at T1.5 (N = 43).

There is an Open Science Framework (OSF) entry for the daily diary study (https://osf.io/kgcdm/) including a detailed description of all obtained measures and this study's ethics protocol. In the current study, we deviated from some of the original questions described on the OSF because the sample size of this study was relatively small. We therefore only described main time effects, and targets and time effects on Dictator Game giving. The hypotheses related to individual differences trajectories should be examined in separate studies with larger sample sizes.

## Materials

**1. Pre-to peri-pandemic changes in mood, empathy, and prosocial behavior.** *Empathy: Empathic concern and perspective-taking*. We measured two components of empathy, perspective-taking and empathic concern, using the respective subscales of the Interpersonal Reactivity Index (IRI; [21]). The perspective-taking subscale included six items (Cronbach's $\alpha_{day1}$ = .79) measuring the inclination to spontaneously adopt the psychological viewpoint of others. A typical item from this scale is "I sometimes try to understand my friends better by imagining how things look from their perspective". The empathic concern subscale included six items (Cronbach's $\alpha_{day1}$ = .75) measuring the tendency to experience feelings of warmth, compassion, and concern for other people. A typical item from this scale is "I often have tender, concerned feelings for people less fortunate than me". Items were rated using a 5-point Likert scale from 0 (does not at all apply to me) to 4 (completely applies to me). For each subscale, the mean of the six items was computed for analyses. The IRI was administered on T1, T2, and day 1, 5, 10, and 15. To increase reliability and as perspective-taking and empathic concern

were expected to provide a personality index, scores on the subscales were averaged over the four daily diary measurements (Becht et al., 2017). As a result, three scores were used for perspective-taking and empathic concern, reflecting T1, T2, and the COVID-19 pandemic.

*General contribution to society.* To measure individual differences in perceived contributions to society, we created a new three-item questionnaire called 'General Contribution to Society' (GCS; Cronbach's $\alpha_{day1}$ = .64). Participants rated on a 10-point scale whether three statements applied to them, with 1 meaning 'not at all' and 10 meaning 'very much'. Before answering the statements, it was explained to participants that contributions to society can take many different forms, e.g. through work, volunteering, or social contacts. The statements were "I think it is important to contribute to society", "I think my volunteering is important", and "I think it is important to make an effort for the people around me". A mean GCS-score was computed by averaging over the three items. Participants were given the option to respond to each of the statements with 'Not applicable', in which case their response was recorded as 'missing' and not used in the mean score computation. The GCS was administered on T2 and day 1, 5, 10, and 15 of the daily diary study. To increase reliability, the GCS-score was averaged over the four daily diaries measurements. As a result, there were two scores for GCS, reflecting T2 and the COVID-19 pandemic.

*Opportunities for prosocial actions*: *Emotional support.* To measure opportunities for prosocial actions (specifically towards friends), we adapted three questions of the 'Opportunities for Prosocial Actions Scale' [22]. Participants rated on a scale from 1 (not at all) to 6 (very much) whether the following three statements applied to them: 'I comforted a friend last day', 'Last day, I did my best to carve out time for friends', and 'Last day, I sent a message to a friend', Cronbach's $\alpha_{day1}$ = .66. The questionnaire was administered on all 15 days of the daily diary study, and in adjusted form on T1, T1.5, T2 (i.e., enquiring the frequency of these actions in the past months, on a scale of 1 (not something I do) to 6 (very often). A mean score for each assessment was computed by averaging over the three items. To increase reliability, scores were averaged per week of the daily diary study and for the five questionnaires obtained at T1.5. As a result, we used six scores for opportunities for prosocial actions, reflecting T1, T1.5, T2, and week 1, 2, and 3 of the daily diary study. A within-person standard deviation was calculated for week 1, 2, and 3 to assess variability in opportunities for prosocial actions during the pandemic (Becht et al., 2017).

*Mood*: *Vigor and tension.* To measure mean levels and fluctuations in vigor and tension, we used the Profiles of Mood States (POMS; [23]). Although the original questionnaire includes more emotions than vigor and tension, these were not measured during the daily diary study (i.e., at T1.5 we did include all emotions as measured by the POMS). We decided to focus only on vigor and tension subscales in the daily diary study to shorten this study's length and minimize pressure for participants. We included a positive emotion (vigor) to get an indication of adolescent resiliency, and a negative emotion (tension) as prior studies have suggested increases in adolescent tension and anxiety as a result of the pandemic [2]. Participants rated the extent to which items represented their mood at that moment, using a 5-point scale ranging from 1 (not at all) to 5 (very well). Scores were calculated by averaging the items for the vigor (5 items; Cronbach's $\alpha_{day1}$ = .71, example item 'active') and tension subscales (6 items; Cronbach's $\alpha_{day1}$ = .90, example item 'anxious'). The POMS was administered on T1.5 and all 15 days of the daily diary study. To increase reliability, the mood and vigor scores were averaged for the five questionnaires obtained at T1.5 and per week of the daily diary study. As a result, we used four mean level scores for vigor and tensions, reflecting T1.5, and week 1, 2, and 3 of the daily diary study. Within-person standard deviations were calculated for week 1, 2, and 3 to assess variability in vigor and tension during the pandemic.

*Prosocial tendencies measure revised*: *Dire and altruistic prosociality*. To measure dire and altruistic prosociality, we used the dire and altruistic subscales of the Prosocial Tendencies Measure Revised (PTM-R; [24]). Dire prosociality refers to helping others in crisis or emergency situations (3 items, Cronbach's $\alpha_{day1}$ = .83, example item: I tend to help people who are in real crisis or need), whereas altruistic prosociality (6 items, Cronbach's $\alpha_{day1}$ = .80, example item: I often help even if I don't think I will get anything out of helping) refers to helping others when there is little or no perceived potential for a direct, explicit reward to the self [24]. Participant rated the extent to which statements applied to them, using a 5-point scale ranging from 1 (not at all) to 5 (very well). Scores were calculated by averaging the items for each subscale, after recoding 5 items on which higher scores reflected less altruistic prosociality. The PTM-R subscales were assessed at T1, T2, and all days of the pandemic daily diary study. To increase reliability, dire and altruistic prosociality scores were averaged per week of the daily diary study. As a result, we used five mean level scores for dire and altruistic prosociality, reflecting T1, T2, and week 1, 2, and 3 of the daily diary study. A within-person standard deviation was calculated for week 1, 2, and 3 to assess variability in dire and altruistic prosociality during the pandemic.

*Social value orientation*: *Slider*. To measure the magnitude of the concern that adolescents have for others, we used the Social Value Orientation (SVO) slider measure, which yields a continuous, non-categorical SVO score [25]. This score is based on six primary SVO slider items and represents the balance between allocations for oneself and others. For each question, adolescents had to decide between nine different outcome allocations for themselves and unfamiliar others, with differing levels of self-gain and other-gain. Higher scores indicate greater prosocial value orientation, with scores greater than 57.15 reflecting an altruistic SVO, scores between 22.45–57.15 reflecting a prosocial SVO, scores between -12.04 and 22.45 reflecting an individualistic SVO, and scores below -12.04 reflecting a competitive SVO. Prior studies have presented evidence for the validity and reliability of the measurement [25].

**2. Peri-pandemic familiarity-, need-, and deservedness-effects on giving.** *Giving to different targets*: *Dictator Games*. Giving behavior was measured on day 1 and day 15 with a novel Dictator Game [5, 16] developed for the purpose of this study, with five trials presented in randomized order. On each trial, participants divided 10 coins between themselves and one of five targets: an unfamiliar peer, a friend, an individual with a poor immune system, an individual with COVID-19, and a doctor working at a hospital. The exact identity of the target remained unknown to participants, they only knew on each trial whether the target was an unfamiliar peer, a friend, etc. Giving behavior was operationalized as the number of donated coins, ensuing in a discrete value between 0 and 10 for each target. The target could not decline the offer or influence it in any way [5]. Participants were instructed to treat the coins as valuable for oneself and the target, and that keeping more coins for oneself would result in fewer coins for the target. To minimize experimenter demand effects, it was emphasized that there were no right or wrong answers, and each trial (i.e., the donation to each target) was shown on a separate screen. Participants were explained that divisions were hypothetical and that there was no actual payment.

## Results

See Table 1 for descriptive statistics and psychometric properties of all measures. Details on how statistical analyses were performed, including data checks, can be found in S1 File. We performed a few descriptive analyses on measures of which we had no pre-pandemic measurements that are beyond the scope of this article. For completeness, these are reported in S2 File, and all measures obtained during the pandemic are described on the Open Science Framework

(https://osf.io/kgcdm/). Histograms of all key variables are displayed in S3 File. An overview and copies of all developed and used questionnaires in English and Dutch can be found in S4 and S5 Files.

## 1. Pre-to peri-pandemic changes in mood, empathy, and prosocial behavior

To examine whether participants had similar levels of empathy (i.e. perspective-taking and empathic concern), general contributions to society, opportunities for prosocial actions, mood (i.e., vigor and tension), dire and altruistic prosociality, and SVO prior to and during the COVID-19 pandemic, Generalized Estimated Equations (GEE) models were performed separately for each variable. GEE models are suitable for longitudinal analyses in small samples as this technique does not apply listwise deletion, estimates robust standard errors, and can accommodate alternative correlation structures (e.g., residual dependency) that are common in repeated measures studies [26]. For perspective-taking an unstructured correlation matrix (i.e., freely estimated correlation matrix) had the best fit (see S6 File; [26]). The Wald Chi-Square test indicated a main effect of time, $X^2$ (2, $N$ = 53) = 12.95, $p$ = .002. Pairwise comparisons with sequential Bonferroni corrections showed that adolescents displayed significantly higher levels of perspective-taking during the COVID-19 pandemic ($M$ = 2.74, $SE$ = .08), compared to T1 ($M$ = 2.51, $SE$ = .08, $p$ = .002) and T2 ($M$ = 2.58, $SE$ = .08, $p$ = .012), see Fig 2. For empathic concern a autoregressive correlation matrix had the best fit, see S6 File. The analysis showed a main effect of time, $X^2$ (2, $N$ = 53) = 34.08, $p$ < .001. Pairwise comparisons with sequential Bonferroni corrections showed that adolescents showed significantly lower levels of empathic concern during the COVID-19 pandemic ($M$ = 2.31, $SE$ = .06), compared to T1 ($M$ = 2.68, $SE$ = .08, $p$ < .001) and T2 ($M$ = 2.64, $SE$ = .09), see Fig 2. No time effect was found for general contributions to society, $p$ = .704.

For mean levels of opportunities for prosocial actions an independent correlation structure had the best fit, see S6 File. The analysis showed a main effect of time, $X^2$ (5, $N$ = 53) = 39.27, $p$ < .001. Pairwise comparisons with sequential Bonferroni corrections showed that adolescents displayed lower levels of opportunities for prosocial actions during week 1 ($M$ = 3.03, $SE$ = .16), week 2 ($M$ = 2.80, $SE$ = .21), and week 3 ($M$ = 2.96, $SE$ = .19) of the pandemic compared to T1 ($M$ = 3.94, $SE$ = .16, $p$'s < .001), and T1.5 ($M$ = 3.76, $SE$ = .15, $p$'s < .001), and T2 ($M$ = 3.55, $SE$ = .19, $p$'s < .008), see Fig 3.

The following analyses focused on POMS mood measures vigor and tension. For vigor an exchangeable correlation structure had the best fit, see S6 File. The analysis showed a main effect of time, $X^2$ (3, $N$ = 53) = 19.58, $p$ < .001. Pairwise comparisons with sequential Bonferroni corrections showed that adolescents displayed higher levels of vigor during week 1 ($M$ = 3.11, $SE$ = .11), week 2 ($M$ = 3.31, $SE$ = .12), and week 3 ($M$ = 3.26, $SE$ = .13) of the pandemic compared to T1.5 ($M$ = 2.79, $SE$ = .08, $p$'s < .006), see Fig 4. For tension an unstructured correlation matrix had the best fit, see S6 File. The analysis showed a main effect of time, $X^2$ (3, $N$ = 53) = 56.65, $p$ < .001. Pairwise comparisons with sequential Bonferroni corrections showed that adolescents displayed lower levels of tension during week 1 ($M$ = 1.77, $SE$ = .10), week 2 ($M$ = 1.47, $SE$ = .07), and week 3 ($M$ = 1.63, $SE$ = .10) of the pandemic compared to T1.5 ($M$ = 2.10, $SE$ = .12, $p$'s < .019), see Fig 4.

GEE models investigating time effects on dire and altruistic prosociality and social value orientation showed no effects of time, $p_{dire}$ = .350, $p_{altruistic}$ = .231, $p_{svo}$ = .612, suggesting no pre- to peri-pandemic changes.

To compare variability levels of opportunities for prosocial actions, vigor, tension, and dire and altruistic prosociality throughout week 1, 2, and 3 of the daily diary study, GEE models were performed. There were no time effects for these variables., suggesting that variability

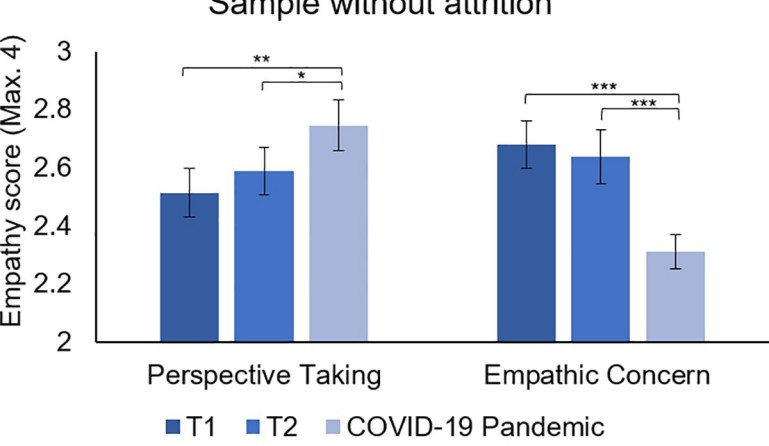

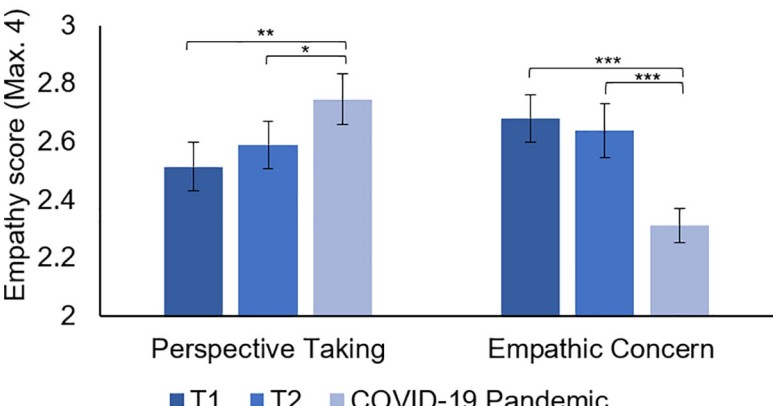

**Fig 2. Levels of perspective-taking and empathic concern at T1, T2, and during the COVID-19 pandemic.**
Adolescents showed higher levels of perspective-taking but lower levels of empathic concern during the COVID-19 pandemic compared to before. Results are displayed separately for the sample without attrition (i.e., consisting of individuals who show no attrition over time, N = 36) and the sample with attrition (i.e., consisting of all available data per time point, regardless of whether participants showed attrition, N = 36 to 53).

levels did not change throughout the daily diaries study, $p_{opportunities\_for\_prosocial\_actions}$ = .085, $p_{vigor}$ = .292, $p_{tension}$ = .436.

Note that this section included multiple tests, which could lead to Type I errors. Therefore, we used Bonferroni correction for multiple comparisons for main effects adjusting for correlating variables [27, 28]. In this section we performed 12 GEE models on variables with an average correlation of .18, which resulted in an adjusted significance level of $\alpha$ = .0042. As such, all aforementioned main effects of time survived this correction. Finally, none of the measures in this section correlated with age, suggesting that the time-related effects were not influenced by age differences.

## 2. Peri-pandemic familiarity-, need-, and deservedness-effects on giving

**Giving toward different targets during the COVID-19 pandemic.** To examine differences in giving towards the five targets on day 1, a GEE model was performed using the five targets as a factor (i.e., an unfamiliar peer, friend, individual with poor immune system, individual with COVID-19, and a doctor working in a hospital). There was no difference between

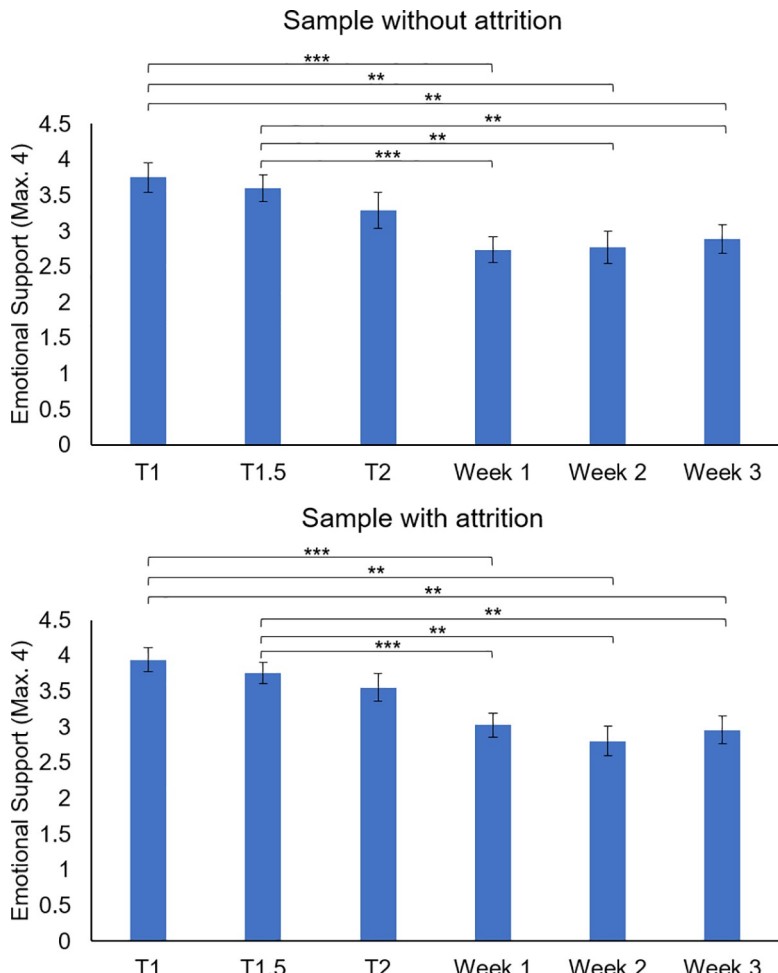

**Fig 3. Levels of opportunities for prosocial actions (emotional support) prior to and during the COVID-19 pandemic.** Timing pre-pandemic measures: T1: May–October 2018; T1.5: between T1 and T2; and T2: August 2019 – January 2020; pandemic measures (i.e., week 1, 2, and 3): March 30 –April 17, 2020. Results are displayed separately for the sample without attrition (i.e., consisting of individuals who show no attrition over time, N = 36) and the sample with attrition (i.e., consisting of all available data per time point, regardless of whether participants showed attrition, N = 36 to 53).

models with different correlation matrices, and therefore results obtained with an independent correlation matrix are being reported here, see S6 File. The analysis showed a main effect of target, $X^2$ (4, N = 53) = 169.70, $p < .001$. Pairwise comparisons with sequential Bonferroni corrections showed that the number of donated coins differed between all targets, $p$'s $\leq .001$. As can be seen in Fig 5, participants gave most coins to a doctor working in a hospital, followed by an individual with COVID-19, followed by an individual with a poor immune system, followed by a friend, and gave the least to an unfamiliar other. To examine whether giving on day 1 differed from giving on day 15, time was added as a factor to the GEE, but the best fitting model revealed no main effect of time, nor an interaction between time and target, $p_{itmen} = .170$ $p_{interaction} = .126$. As can be seen in Fig 5, giving patterns were highly similar for day 1 and day 15 for all targets. Table 2 shows significant correlations in giving to each target across the two measurements. As can be seen in Table 2, only giving to a friend on day 15 was negatively correlated with age, such that young adolescents gave more to friends than older adolescents. None of the other correlations with age were significant.

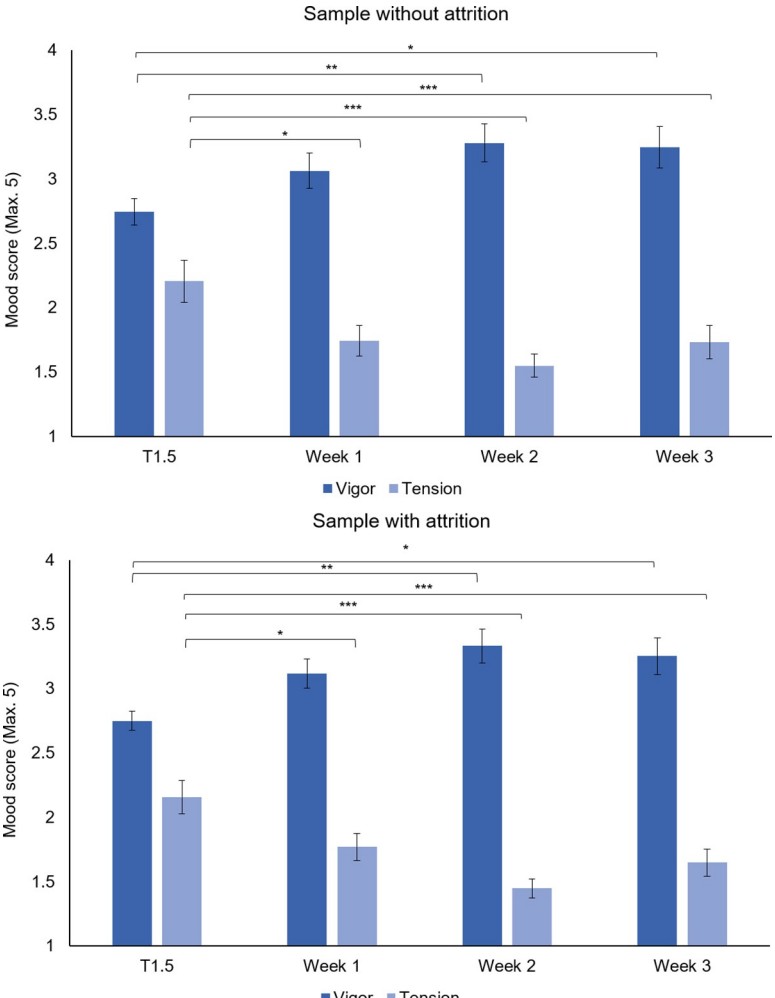

**Fig 4. Levels of vigor and tension prior to and during the COVID-19 pandemic.** Timing pre-pandemic measures: 2018 –January 2020; pandemic measures (i.e., week 1, 2, and 3): March 30 –April 17, 2020. Results are displayed separately for the sample without attrition (i.e., consisting of individuals who show no attrition over time; N = 36) and the sample with attrition (i.e., consisting of all available data per time point, regardless of whether participants showed attrition, N = 36 to 53).

In this section, 53 correlations and 2 GEE models were calculated for variables with an average correlation of .14. Using Bonferroni correction for multiple comparisons for main effects adjusting for correlating variables [27, 28], this resulted in an adjusted significance level of α = .0009, which only the analysis examining the main effect of target survived. Note that p-values below .001 were reported as $p < .001$. Correlations and associated significance levels are still reported in Table 2 to inform future research.

## Discussion

In the current study we investigated: 1) effects of the COVID-19 pandemic and the associated lockdown on Dutch adolescents' mood, empathy, and prosocial behavior, and 2) familiarity-, need-, and deservedness-effects on peri-pandemic Dictator Game giving. Results related to the first part showed that during three of the first weeks of pandemic lockdown, adolescents across ages 10–20 years showed increased levels of social-cognitive perspective-taking and vigor

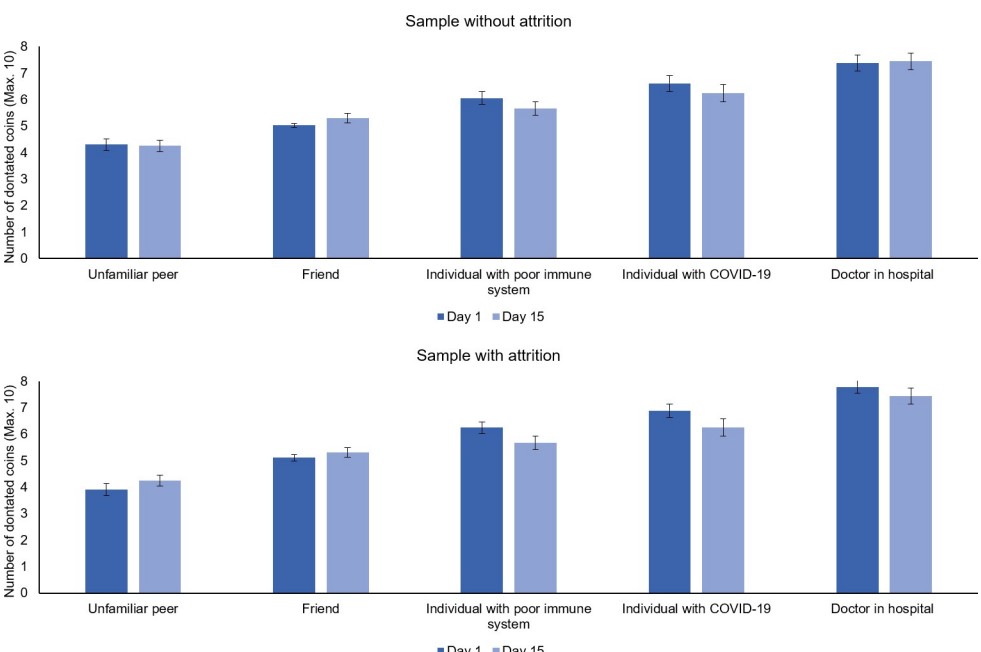

**Fig 5. Giving to an unfamiliar peer, friend, individual with a poor immune system, individual with COVID-19, and doctor working on a hospital.** Giving was assessed on day 1 (i.e., March 30, 2020) and 15 (i.e., April 17, 2020). Giving to all targets differed significantly from each other on day 1, as indicated by Bonferroni corrected pairwise comparisons, p's < .010. There were no differences in giving between day 1 and 15. Results are displayed separately for the sample without attrition (i.e., consisting of individuals who show no attrition over time; N = 36) and the sample with attrition (i.e., consisting of all available data per time point, regardless of whether participants showed attrition, N = 36 to 53).

compared to before the pandemic lockdown. Furthermore, they showed stable levels of general contributions to society, social value orientation, altruism and dire prosociality. Finally, they showed decreased levels of empathic concern, opportunities for prosocial actions, and tension. Results related to the second part showed that adolescents showed higher levels of peri-pandemic giving to a friend (i.e., familiar other; 51% of resources) relative to an unfamiliar peer (39% of resources). Adolescents also showed higher levels of giving to a doctor in the hospital (deserving, 78% of resources) and individuals with COVID-19 or a poor immune system (in need; i.e., 69 and 63% of resources, respectively) relative to an unfamiliar peer. The discussion is organized in line with these findings.

## Pre- to peri-pandemic changes in mood, empathy, and prosocial behavior

The findings of this study suggest that the COVID-19 pandemic influences adolescents' mood and social inclinations. Adolescents showed relatively lower levels of empathic concern during the first weeks of lockdown, which prior studies already have shown to be an important predictor for social relations [19]. Lower levels of empathic concern during the pandemic relative to measures in the same individuals obtained one and two years prior could indicate that adolescents became relatively more emotionally self-focused when experiencing lockdown and social distancing [1]. Indeed, opportunities for prosocial actions also declined relative to measurements prior to the pandemic. This suggests that adolescents had fewer opportunities to feel with and support others due to their limited opportunities to interact with people outside their household [2, 29]. Although studies examining levels of empathic concern in crisis situations are scarce, especially with regard to adolescence, studies in adults have shown that social

**Table 2. Pearson correlations between age and giving to different targets on day 1 and 15, as measured during the daily diaries study.**

| | Giving scores day 1 (N = 53) | | | | | Giving scores day 15 (N = 36) | | | | |
| --- | --- | --- | --- | --- | --- | --- | --- | --- | --- | --- |
| | Unfamiliar peer | Friend | Individual with poor immune system | Individual with COVID-19 | Doctor working at a hospital | Unfamiliar peer | Friend | Individual with poor immune system | Individual with COVID-19 | Doctor working at a hospital |
| Age (N = 53) | .13 | -.22 | -.06 | .00 | -.15 | -.18 | -.39* | .04 | -.09 | .10 |
| Giving scores day 1 (N = 53) | | | | | | | | | | |
| Unfamiliar Peer | – | | | | | | | | | |
| Friend | -.09 | – | | | | | | | | |
| Individual with poor immune system | .06 | -.15 | – | | | | | | | |
| Individual with COVID-19 | .05 | .17 | .72** | – | | | | | | |
| Doctor working at a hospital | .03 | .15 | .37** | .46** | – | | | | | |
| Giving scores day 15 (N = 36) | | | | | | | | | | |
| Unfamiliar Peer | .19** | .04 | .16 | .19 | .17 | – | | | | |
| Friend | -.36* | .28 | -.12 | -.04 | .10 | .30 | – | | | |
| Individual with poor immune system | .42** | -.20 | .36* | .42* | .45** | .34* | -.11 | – | | |
| Individual with COVID-19 | .27 | -.14 | .40* | .57** | .42* | .23 | -.18 | .74** | – | |
| Doctor working at a hospital | .12 | -.19 | .24 | .10 | .54** | -.05 | -.13 | .58** | .44** | – |

Note: $p \leq .050$

**: $p \leq .010$

***: $p \leq .001$.

interactions and support are important predictors of resilience and a better emergence from crisis situations [30]. Thus, a larger body of evidence has shown that social interactions are necessary for empathic social development. Therefore, future studies should examine long-term socio-emotional consequences of the reported decreases in empathic concern and opportunities for prosocial actions.

Interestingly, adolescents showed increases in social-cognitive perspective-taking and vigor, and decreases in tension relative to time points prior to the pandemic. The increase in vigor and decrease in tension was unexpected, but might indicate that adolescents normally experience stress and pressure, possibly related to school or other obligations, which has been shown to be a growing problem for young people [31, 32]. These findings are in line with prior studies showing that crisis situations can sometimes have positive effects on mental health and personal growth [30, 33]. It should be noted that our findings could be specific for adolescents and young adults [34], as a recent longitudinal study with pre-pandemic data demonstrated that adults show increases in depression and anxiety during the pandemic [35]. The increase in perspective-taking might indicate that the crisis offered adolescents an opportunity to cognitively empathize with others. In prior studies higher levels of perspective-taking have been associated with increased giving to and willingness to interact with out-group members [5, 36]. Future research should examine whether any beneficial effects persist during and after the COVID-19 pandemic and whether they have long-term effects on adolescents' social behavior.

In contrast to opportunities for prosocial actions, which declined during the pandemic, we found no pre- to peri-pandemic changes in general contributions to society, social value orientation and altruistic and dire prosociality. Our findings suggest no changes in general prosocial tendencies towards unfamiliar others or society as a whole, only a decline in concrete daily prosocial actions towards familiar others (i.e., friends). Interestingly, recent studies in adults have suggested that during the pandemic there is a shift towards pandemic-specific prosocial behaviors. For example, these studies have demonstrated that the pandemic makes individuals more likely to prioritize society's problems over their own, but less willing to give to non-COVID-19 related causes [37, 38]. This interpretation also fits with our finding that Dictator Game giving is higher for COVID-19 related targets. Future studies should aim to replicate this finding in both adults and adolescents and examine whether this shift persists further into and after the pandemic.

## Peri-pandemic giving to others

The finding that adolescents give more to familiar others, and those who are in need or deserving, is in line with prior studies [5, 7, 15]. Here, adolescents gave significantly more to friends (i.e., around 50%; [5, 15]) than to unfamiliar peers. Secondly, we found that adolescents donated more to doctors (individuals who are deserving), individuals with COVID-19, and individuals with a poor immune system (individuals in need in the context of the COVID-19 pandemic) compared to friends and unfamiliar peers. This suggests that need and deservedness had a greater influence on adolescent giving than familiarity in the ecologically valid context of the COVID-19 pandemic. Future studies should investigate whether the high levels of giving to COVID-19 related targets are long-term or wear off over time. Another interesting direction for future research is to link these types of giving with COVID-19 related health behaviors and engagement with government regulations. Recent studies in adults have namely demonstrated links between pandemic health behaviors and various forms of prosociality [39–42]. Finally, the novel Dictator Games provided a reliable measure of giving as was shown by significant correlations across time (i.e., $r$'s .19-.57), although these correlations were lower than those of studies where Dictator Games to anonymous others were repeatedly administered in the same sitting (i.e., $r$'s .70-.91). The only target for which giving was not significantly correlated over time was the friend, which was also the target that showed an association with age at the final time point. Possible explanations include that adolescents thought of a different friend across time points or that their relationship changed over the course of three weeks. The correlations over time for the other targets may also have been influenced by time-related variability in target perception, especially since these measures were taken in the first weeks of the pandemic [43].

## Strengths, limitations, and recommendations for future research

This study had several strengths. First, we used intensive daily diary assessments during the pandemic. As such, measurements within individuals were based upon multiple assessments which increases robustness and reliability. Second, to draw reliable conclusions about changes in behavior, it is important that measurements are obtained pre- and peri-pandemic, as was the case in our longitudinal project. The within-subject-, longitudinal design enhanced statistical power. Third, this study obtained data in the first weeks of the pandemic, and is therefore unique in capturing the very early experiences of being in lockdown. Fourth, the novel experimental Dictator Games allowed us to disentangle familiarity-, need-, and deservedness-effects on giving in an ecological valid context (i.e., the COVID-19 pandemic). Fifth, this study

focused on pandemic effects on adolescents, a relatively understudied age period that is formative for socio-emotional development.

The current study also has several limitations that should be addressed in future research. First, to the awareness of participants the Dictator Games and associated payout were hypothetical, which could raise concerns regarding ecological validity. Reassuringly, as supported by meta-analytic evidence, individuals generally give away a similar number of resources in Dictator Games when real money is involved [20]. Several studies demonstrated associations between economic games such as the Dictator Game and actual and self-reported prosocial behaviors [5, 44, 45]. However, some studies failed to find such associations [46], and therefore more research is needed to evaluate the external validity of Dictator Games, including variations with multiple targets. Second, we invited adolescents enrolled in a cohort-sequential longitudinal study to participate in the daily diary study during the pandemic. While this allowed us to compare experiences and social behavior prior to and during the COVID-19 pandemic, it could have introduced self-selection bias (e.g. inclusion of empathic and altruistic individuals) and it limited our potential sample size. There are several factors that could explain the attrition in this study. First, the pandemic and lockdown resulted in many changes in the lives of adolescents and their parents, including home schooling, new family routines and online communication. The current study was performed ad hoc and required participants to sign up within the period of one week. This may have caused larger attrition than is typical in longitudinal studies. Second, it is noteworthy that attrition was larger for males than for females. There is currently no explanation for this difference, but it is worth noting that other pandemic studies have also shown higher participation of females compared to males [34, 35, 37, 39]. To investigate the extent to which self-selection bias influenced our results, we examined whether adolescents who did not participate in the daily diary study differed from those who did, see S7 File. These analyses showed that adolescents who participated in the daily diary study showed slightly higher pre-pandemic levels of empathic concern and altruism, but did not differ in pre-pandemic levels of mood, perspective-taking, age, and dire prosociality. Future studies should employ large, diverse samples to test the robustness of our findings and to test for individual differences (e.g., in giving). A third limitation is that the COVID-19 pandemic might impact the saliency of other emotions than tension and vigor, such as fear (e.g., fear of death or to lose a loved one), which in turn can affect certain forms of prosociality [12]. Future studies should investigate the effect the COVID-19 pandemic on such emotions and assess their relationship with prosocial tendencies. Fourth, we cannot conclude that changes in mood and social inclinations reported here were only driven by the pandemic. Time trends such as aging and seasonality may also play a role. However, given the relatively large changes between the second measurement wave and pandemic—despite the short time in between–we deem it most plausible that effects are directly related to the pandemic. Fifth, another direction for future research includes social media and other forms of digital social interactions, and how they influence the interplay between adolescents' social behavior and wellbeing during the COVID-19 pandemic. Finally, this study only addressed immediate experiences in the first weeks of lockdown. Future studies should examine the long-term effects of the pandemic and social distancing [2].

## Conclusion

In conclusion, this daily diary study used a cohort-sequential sample of adolescents to examine pre- to peri-COVID 19 pandemic changes in mood, empathy, prosocial behavior, as well as familiarity-, need- and deservedness- effects on Dictator Game giving. We demonstrated that levels of empathic concern and opportunities for prosocial actions dropped during the

COVID-19 pandemic, possibly reflecting a lack of perceived opportunity due to social distancing measures [2]. However, adolescents also showed resilience or even possible benefits as a result of the lockdown, as evidenced by increased levels of perspective-taking and vigor, decreased levels of tension, and high levels of giving to COVID-19 related targets. The results of this study shed new light on adolescence as a crucial period in life for social interactions, and as a phase that is not only characterized by risk behavior, but also by resilience and a willingness to meaningfully contribute to others and society [7, 8]. It will be an important aim for researchers and policy makers to utilize this feature of adolescence, for example by involving adolescents in efforts to shape society and the future for the better. Finally, this study demonstrated that various types of prosociality were differentially impacted by the COVID-19 pandemic, possibly as a result of a shift towards pandemic-specific prosocial behaviors. This result highlights the need to differentiate between various forms of prosocial behaviors to increase our understanding of their etiology and how they shape human development.

## Supporting information

**S1 Fig. Age and gender distributions of the daily diary sample.**
(TIF)

**S1 File. Details regarding statistical analyses, including assumption checks.**
(DOCX)

**S2 File. Results for other measures assessed during the daily diary study.**
(DOCX)

**S3 File. Histograms of key variables.**
(DOCX)

**S4 File. Overview and copies of all used and developed questionnaires in English.**
(DOCX)

**S5 File. Overview and copies of all used and developed questionnaires in Dutch.**
(DOCX)

**S6 File. Details on GEE analyses.**
(DOCX)

**S7 File. Attrition: Did participants who decided not to participate in the daily diary study differ from those who did participate?.**
(DOCX)

## Acknowledgments

We would like to thank all participants for their participation.

## Author Contributions

**Conceptualization:** Suzanne van de Groep, Kiki Zanolie, Kayla H. Green, Sophie W. Sweijen, Eveline A. Crone.

**Data curation:** Suzanne van de Groep.

**Formal analysis:** Suzanne van de Groep, Eveline A. Crone.

**Funding acquisition:** Eveline A. Crone.

**Investigation:** Suzanne van de Groep, Kayla H. Green, Sophie W. Sweijen.

**Methodology:** Suzanne van de Groep, Kiki Zanolie, Kayla H. Green, Sophie W. Sweijen, Eveline A. Crone.

**Project administration:** Suzanne van de Groep, Kiki Zanolie, Kayla H. Green, Sophie W. Sweijen.

**Supervision:** Eveline A. Crone.

**Validation:** Eveline A. Crone.

**Visualization:** Suzanne van de Groep.

**Writing – original draft:** Suzanne van de Groep.

**Writing – review & editing:** Suzanne van de Groep, Kiki Zanolie, Kayla H. Green, Sophie W. Sweijen, Eveline A. Crone.

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
