## [Decision Letter · Decision Letter 0]

16 Jul 2020

PONE-D-20-19516

A daily diary study on adolescents’ mood, concern for others, and giving behavior during the COVID-19 pandemic

PLOS ONE

Dear Dr. van de Groep,

Thank you for submitting your manuscript to PLOS ONE. After careful consideration, we feel that it has merit but does not fully meet PLOS ONE’s publication criteria as it currently stands. Therefore, we invite you to submit a revised version of the manuscript that addresses the points raised during the review process.

Please find below the reviewers' and mine's comments.

We look forward to receiving your revised manuscript.

Kind regards,

Valerio Capraro

Academic Editor

PLOS ONE

Journal Requirements:

Additional Editor Comments (if provided):

I have now collected two reviews from two experts in the field. Both reviews are positive but suggest substantial revisions. Therefore, I would like to invite you to revise your work following the reviewers' comments. Additionally, I would like to add a few more comments. (i) in the discussion, you say that dictator game giving when real money is involved is higher than hypothetical giving, and you cite a paper by Engel. First of all, Engel's paper has been published in 2011 in Experimental Economics (please update your reference list). But, more importantly, your statement is wrong: Engel reports no statistically significant difference between hypothetical and real decisions. (ii) You also say that several studies have found correlation between dictator game giving and actual altruism; but you fail to mention that there are also several studies who did not find any, as for example Galizzi & Navarro-Martinez's meta-analysis. (iii) in the general introduction, the "perspective article" about what social and behavioural science can do to promote COVID-19 pandemic response, published by Van Bavel et al. on Nature Human Behaviour, can be a useful general reference. (iv) one might argue that the pandemic has made salient some emotions in general and fear of death in particular. We know from previous work that mortality salience has an effect on pro-social behaviour and that emotions in general have an effect on prosocial behaviour (see Capraro 2019 for a review). Perhaps it would be useful to discuss potential links between your work and this of work.

Of course, it is not a requirement to cite all these works, but I am mentioning them because they seem very related to your work.

I am looking forward for the revision.

References:

Capraro, V. (2019). The dual-process approach to human sociality: A review. Available at SSRN 3409146.

Galizzi, M. M., & Navarro-Martinez, D. (2019). On the external validity of social preference games: a systematic lab-field study. Management Science, 65, 976-1002.

Van Bavel, J. J., et al. (2020). Using social and behavioural science to support COVID-19 pandemic response. Nature Human Behaviour.

Reviewers' comments:

Reviewer's Responses to Questions

**Comments to the Author**

1. Is the manuscript technically sound, and do the data support the conclusions?

Reviewer #1: Partly

Reviewer #2: Yes

2. Has the statistical analysis been performed appropriately and rigorously? 

Reviewer #1: No

Reviewer #2: N/A

3. Have the authors made all data underlying the findings in their manuscript fully available?

Reviewer #1: No

Reviewer #2: Yes

4. Is the manuscript presented in an intelligible fashion and written in standard English?

Reviewer #1: Yes

Reviewer #2: Yes

5. Review Comments to the Author

Reviewer #1: Summary: This paper investigates how the COVID-19 pandemic lockdown affects social concerns and mood of adolescents by comparing measures before and during the pandemic. Moreover, the authors study whether dictator giving towards people associated with COVID-19 (doctors, individual with COVID-19, individual with a poor immune system) differs from dictator giving towards others (friend, unfamiliar peer). Finally, they correlate dictator giving with different measures of empathy and social concerns.

Evaluation: I think it is important to study how COVID-19 policies affect development of social preferences among adolescents. However, I am concerned about the robustness and validity of the findings presented in the paper: there is no effect for many outcome variables, the sample size is small and the research questions differ substantially from the pre-registration plan. I think the paper should be revised substantially to discuss these concerns and do additional robustness tests.

Comments

1. No effects on many outcomes variables

A key question addressed in this project is how the COVID-19 pandemic affects concerns for others and social behavior. In paper, authors report effects on emphatic concerns, perspective taking, general contribution to society, opportunities for prosocial actions and mood. However, the authors elicit additional measures of social behavior such as dire, altruism and social value orientation. They report results for these measures in Appendix S2:

“We examined whether adolescents’ levels of dire and altruistic prosociality, as well as their social value orientation changed when comparing measures prior to and during the pandemic. Repeated measures ANOVAs indicated no changes over time in these variables.”

I think these null results should be reported in the main text of the paper. This allows the reader to assess the validity of the findings. Moreover, given that the authors look at many outcome variables and only find statistically significant effects for some of them, there seems to be a potential problem with multiple hypothesis testing. I think the authors should account for this issue in the paper.

2. Could results be driven by a time trend?

The authors compares concerns for others before and during the pandemic. While it seems likely that the reported effects are driven by the experience of the pandemic, they could also come from a time trend (e.g., aging or seasonality). While the authors can not control for time trends, I think they should discuss this concern in the paper (for example, by arguing that we would not expect any aging effect to occur during such a short time span).

3. Small sample size

Sample sizes are in between 36 and 53, which makes me worried that the study might be underpowered.

4. Potential for experimenter demand effects

The authors find that participants give more money to others if they are associated with COVID-19 (in hypothetical choices). I am worried that this effect is driven by experimenter demand effects. This concern might be partially addressed by controlling for the social desirability measure they collect. (Although social desirability might also reflect prosociality).

5. Research questions differ from pre-registration

The research questions studied seem to differ substantially from what was pre-registered (https://osf.io/kgcdm/). I think the authors should mention this in the paper and report the results for the initial questions in the Appendix.

6. I think the writing of the paper could be improved

I think the writing of the paper could benefit from a stronger focus on its core contribution and from pointing out what its key insights and implications are. (For example, it is not clear to me what we learn from dictator giving towards people associated with COVID-19.)

Reviewer #2: Overview

The study makes multiple, tentatively related observations using a unique survey of adolescents. The authors present one set of findings comparing pre-pandemic and pandemic data: mood improves (vigor increases, tension decreases), empathetic concern decreases, and perspective taking increases. Drawing on data exclusively collected during the pandemic the authors furthermore observe that adolescents state that they would give the same amount of money to healthcare workers as to friends. The authors argue that these results are intimately tied to reduced social contacts during the pandemic. While I think that studying the impact of the pandemic on adolescents is important, I have some concerns which I detail below.

Major Concerns:

1) Connection between motivation and results

a. The motivation is focuses heavily on social contacts which is one important thing that changed during lockdown. But a lot of other important things changed as well, which may have/did affect mood, worries about the future, worries about the health of family members etc. The described link between social contacts and the behaviors investigated is too tentative for me. The authors should give more room to other factors that could also play a role in the introduction. This would also allow them to describe more nuanced expectations (see below).

6. PLOS authors have the option to publish the peer review history of their article (what does this mean?). If published, this will include your full peer review and any attached files.

Reviewer #1: No

Reviewer #2: No

---

## [Author Response · Author response to Decision Letter 0]

14 Aug 2020

We thank the editor for the opportunity to revise our article ‘A daily diary study on adolescents’ mood, concern for others, and giving behavior during the COVID-19 pandemic. Please find below our replies to the helpful comments we received. 

Additional Requirements

Answer: We checked whether our manuscript meets all PLOS ONE’s style requirements and made adjustments where needed. 

Answer: We included copies of all used and developed questionnaires in both Dutch and English in S5 and S6. Note that this information is also available online: https://osf.io/kgcdm/. 

Answer: All data is now uploaded to the EUR repository and follows FAIR principles. The data can be found through this confidential link: https://figshare.com/s/4d0529fe38e7baa5e2a8. Data will be made publicly available upon acceptance using DOI 10.25397/eur.12783161. 

Answer: We included captions for all our Supporting Information files at the end of the manuscript, and updated all in-text citations accordingly. 

Additional Editor Comments: 

I have now collected two reviews from two experts in the field. Both reviews are positive but suggest substantial revisions. Therefore, I would like to invite you to revise your work following the reviewers' comments. Additionally, I would like to add a few more comments. (i) in the discussion, you say that dictator game giving when real money is involved is higher than hypothetical giving, and you cite a paper by Engel. First of all, Engel's paper has been published in 2011 in Experimental Economics (please update your reference list). But, more importantly, your statement is wrong: Engel reports no statistically significant difference between hypothetical and real decisions. (ii) You also say that several studies have found correlation between dictator game giving and actual altruism; but you fail to mention that there are also several studies who did not find any, as for example Galizzi & Navarro-Martinez's meta-analysis. (iii) in the general introduction, the "perspective article" about what social and behavioural science can do to promote COVID-19 pandemic response, published by Van Bavel et al. on Nature Human Behaviour, can be a useful general reference. (iv) one might argue that the pandemic has made salient some emotions in general and fear of death in particular. We know from previous work that mortality salience has an effect on pro-social behaviour and that emotions in general have an effect on prosocial behaviour (see Capraro 2019 for a review). Perhaps it would be useful to discuss potential links between your work and this of work.

Of course, it is not a requirement to cite all these works, but I am mentioning them because they seem very related to your work.

I am looking forward for the revision.

References:

Capraro, V. (2019). The dual-process approach to human sociality: A review. Available at SSRN 3409146.

Galizzi, M. M., & Navarro-Martinez, D. (2019). On the external validity of social preference games: a systematic lab-field study. Management Science, 65, 976-1002.

Van Bavel, J. J., et al. (2020). Using social and behavioural science to support COVID-19 pandemic response. Nature Human Behaviour.

Answer: We thank the editor for the opportunity to revise our manuscript. (i) We agree with the editor’s comment regarding the Engel paper, as the effect of handling real money was only visible when observing distributions instead of means. We adjusted our statement regarding the Engel paper as follows on pages 26-27: “Reassuringly, as supported by meta-analytic evidence, individuals generally give away a similar number of resources in Dictator Games when real money is involved (16)”. (ii) Thank you for acquainting us with the paper by Galizzi & Navarro. We now address your point in the discussion on page 27: “However, some studies failed to find such associations (45), therefore more research is needed to evaluate the external validity of Dictator Games, including variations with multiple targets.”. (iii) We agree that the Van Bavel article is a useful general reference and have incorporated it into our manuscript, for example on pages 3-4 “The COVID-19 pandemic in 2020 represents a massive and challenging global health crisis (9). Without a vaccine, most governments have authorized containment measures, such as social distancing, to slow the pandemic (9). However, distancing conflicts with the human tendency to connect with others, which is even more pronounced in adolescence, and severely limits opportunities to have contact with people outside one’s household (1,2,9)”. (iiii) We added a brief discussion of the importance of emotions and specifically fear of death during the COVID-19 pandemic and the possible impact on prosociality on page 27: “Third, the COVID-19 pandemic might influence the saliency of other emotions than tension and vigor, such as fear (e.g., fear of death or to lose a loved one), which in turn can affect certain forms of prosociality (33). Future studies should investigate the effect the COVID-19 pandemic on such emotions and assess their relationship with prosocial tendencies.”

Reviewers' comments:

Reviewer 1

Reviewer #1: Summary: This paper investigates how the COVID-19 pandemic lockdown affects social concerns and mood of adolescents by comparing measures before and during the pandemic. Moreover, the authors study whether dictator giving towards people associated with COVID-19 (doctors, individual with COVID-19, individual with a poor immune system) differs from dictator giving towards others (friend, unfamiliar peer). Finally, they correlate dictator giving with different measures of empathy and social concerns.

Evaluation: I think it is important to study how COVID-19 policies affect development of social preferences among adolescents. However, I am concerned about the robustness and validity of the findings presented in the paper: there is no effect for many outcome variables, the sample size is small and the research questions differ substantially from the pre-registration plan. I think the paper should be revised substantially to discuss these concerns and do additional robustness tests.

Comments

1. No effects on many outcomes variables

A key question addressed in this project is how the COVID-19 pandemic affects concerns for others and social behavior. In paper, authors report effects on emphatic concerns, perspective taking, general contribution to society, opportunities for prosocial actions and mood. However, the authors elicit additional measures of social behavior such as dire, altruism and social value orientation. They report results for these measures in Appendix S2:

“We examined whether adolescents’ levels of dire and altruistic prosociality, as well as their social value orientation changed when comparing measures prior to and during the pandemic. Repeated measures ANOVAs indicated no changes over time in these variables.”

I think these null results should be reported in the main text of the paper. This allows the reader to assess the validity of the findings. Moreover, given that the authors look at many outcome variables and only find statistically significant effects for some of them, there seems to be a potential problem with multiple hypothesis testing. I think the authors should account for this issue in the paper.

Answer: We thank the reviewer for the suggestion to include all pre- and peri-pandemic measures on social behavior in the main text. Therefore, we now included methods and results regarding dire prosociality, altruistic prosociality, and social value orientation, see pages 15-16; 19. We also incorporated interpretations of these measures in the introduction and discussion. We did not include analyses based on other measures that were only obtained peri-pandemic (e.g., risk taking, rumination) and that were beyond the scope of the article in Supplement 2 (S2). This paper focused on emotions and social behavior specifically. 

We agree that it is important to control for Type I errors. Therefore, we included the following paragraph at the end of the method section (p.16): “Note that the present study included multiple tests, which could lead to Type I errors. We used Bonferroni correction for multiple comparisons adjusting for correlating variables (26,27) for main effects pertaining to both research aims. For the first research aim (i.e., investigating pre-to-peri pandemic changes in mood, empathy, and prosocial behavior) we performed 15 Repeated Measures (RM) ANOVAs on variables with an average correlation of .19, which resulted in an adjusted significance level of α = .0056. For the second research aim (i.e., peri-pandemic familiarity-, need-, and deservedness-effects on giving) we performed 105 correlations and 3 RM ANOVA’s on variables with an average correlation of .16, which resulted in an adjusted significance level of α = .0009. To interpret main effects found in RM ANOVAs, post-hoc tests were conducted using Bonferroni corrected pairwise comparisons. As such, most confidence should be placed in the results reported with p‐values < .007 and < .001, respectively (p‐values below .001 were reported as p < .001). Exact p‐values are still reported in cases where p > .001, to inform future research.”

2. Could results be driven by a time trend?

The authors compares concerns for others before and during the pandemic. While it seems likely that the reported effects are driven by the experience of the pandemic, they could also come from a time trend (e.g., aging or seasonality). While the authors can not control for time trends, I think they should discuss this concern in the paper (for example, by arguing that we would not expect any aging effect to occur during such a short time span).

Answer: We agree that we cannot strictly conclude that reported effects with regard to concern for others are only driven by the experience of the pandemic, and not by time trends aging or seasonality. However, Figure 2 shows that the change between T2 and the pandemic is larger than the change between T1 and T2, despite T2 and the pandemic being closer together in time, see below. This makes it unlikely that the results are due to aging. 

T1: May – October 2018

T2: August 2019- January 2020

Pandemic: March/April 2020

We now included this in the discussion on page 27: “Fourth, we cannot conclude that the results regarding changes in mood and social inclinations are only driven by the pandemic. Time trends such as aging and seasonality may also play a role. However, given the large changes between the second measurement wave and pandemic - despite the short time span and in comparison to trends observed prior to the pandemic– we deem it most plausible that effects are directly related to the pandemic."

3. Small sample size

Sample sizes are in between 36 and 53, which makes me worried that the study might be underpowered.

Answer: We agree that studies ideally have larger sample sizes. However, there are several reasons why this study makes a valuable contribution to the literature. First, we used intensive daily diary assessments, therefore measurements within individuals were based on multiple assessments which increases robustness and reliability. Second, to draw reliable conclusions about changes in behavior, it is important that measurements are obtained prior to and during the pandemic, as is the case in our longitudinal project. Within-subject-, longitudinal designs enhance statistical power, which in our view outweighs the disadvantages of the smaller sample size. Third, this study obtained data in the first weeks of the pandemic, and therefore is unique in capturing the very early experiences of being in lockdown. Other studies should replicate the reported effects using larger sample sizes, which we mention in the discussion on page 27: “Secondly, we invited individuals who participated in a cohort-sequential longitudinal study to participate in the daily diary study during the pandemic. While this allowed us to compare experiences and social behavior prior to and during the COVID-19 pandemic, it could also have introduced self-selection bias (e.g. inclusion of empathic and altruistic individuals) and it limited our potential sample size. The attrition was larger in males, a trend that is also observed in other studies (31,32,34,36). Future studies should employ large, diverse samples to test the robustness of these findings and to test for individual differences.” 

4. Potential for experimenter demand effects

The authors find that participants give more money to others if they are associated with COVID-19 (in hypothetical choices). I am worried that this effect is driven by experimenter demand effects. This concern might be partially addressed by controlling for the social desirability measure they collect. (Although social desirability might also reflect prosociality).

Answer: Based on your suggestion, we tested whether the target effects in the Dictator Game still held when controlling for social desirability. We added the social desirability response set (SDRS-5) variable as a covariate to the repeated measures ANOVA within target (5 levels) as a within-subject variable. There was no main effect of social desirability on giving (p = .371), nor was there an interaction between social desirability and target (p = .223). In addition, calculating bivariate correlations between social desirability and giving to each of the five targets revealed no significant associations. 

These findings strengthen our interpretation that the results obtained in this study are not driven by experimenter demand effects. Furthermore, our instructions were carefully designed to prevent experimenter demand effects or norm communication, as participants were explained that there were no right or wrong answers and they could decide upon the division of coins by themselves. Typically, norm driven behavior would lead to equity divisions in zero sum games (Fehr & Fischbacher, 2003; Fehr & Schmidt, 1999; Meuwese, Crone, de Rooij, & Güroglu, 2015), therefore, the larger donations to COVID-19 targets can be interpreted as prosocial behavior. Finally, we displayed each trial (i.e., the donation to each target) on separate screens, which together with the randomized order minimized adjustments based on targets yet to come. The details regarding the instructions are now more explicitly mentioned in the methods (pages 16-17, e.g. “To minimize experimenter demand effects, it was emphasized that there were no right or wrong answers, and each trial (i.e., the donation to each target) was shown on a separate screen.”). 

5. Research questions differ from pre-registration

The research questions studied seem to differ substantially from what was pre-registered (https://osf.io/kgcdm/). I think the authors should mention this in the paper and report the results for the initial questions in the Appendix.

Answer: For the current study, we pre-registered our ethics protocol and all the measurement instruments we used. Several questions from the pre-registered ethics protocol could not be answered in this study due to the small sample size and to prevent further density of the manuscript. For example, the questions “When and how do changes in emotional reactivity result in risks and opportunities for prosocial development and which factors facilitate opportunities for positive, prosocial development?” and “Which experiences at what ages create the greatest opportunities for positive and prosocial development and which experiences trigger self-protection mechanisms?” can best be answered in a study using a larger sample size to successfully account for individual differences. 

We explain this on page 12 of the manuscript: “There is an Open Science Framework (OSF) entry for the daily diary study (https://osf.io/kgcdm/) including a detailed description of all obtained measures and this study’s ethics protocol. In the current study, we deviated from some of the original questions described on the OSF because the sample size of this study was relatively small. We therefore only described main time effects, and explored individual differences in Dictator Game giving. The hypotheses related to individual differences trajectories should be examined in separate studies with larger sample sizes.”

6. I think the writing of the paper could be improved

I think the writing of the paper could benefit from a stronger focus on its core contribution and from pointing out what its key insights and implications are. (For example, it is not clear to me what we learn from dictator giving towards people associated with COVID-19.)

Answer: We made several adjustments to create a stronger focus on our two core contributions: 1) comparing pre-to peri pandemic measures of mood, empathy and prosocial behavior, and 2) to investigate familiarity- need- and deservedness-effects on giving; and to point out key insights and implications. For example, we more explicitly mention the core aims in the abstract, introduction, and discussion, and used sub-headings in the introduction and discussion to further emphasize the two main aims. 

Prior psychological and economic literature shows that ‘need’ and ‘deserving’ are important drivers of prosocial behavior to unfamiliar others (see for example Engel, 2011). The pandemic poses a unique situation as it makes the need of certain targets particularly evident in an ecologically valid context. As prior studies in adolescents have shown the importance of whether a target is unknown or familiar (e.g., an unfamiliar peer vs. friend) for giving behavior, we were interested to extend these findings by including unfamiliar targets that were in need or deserving. For this purpose, we used the validated Dictator Game to examine giving towards people associated with COVID-19. These findings may provide us with more insight into mechanisms and motivations underlying giving to different types of targets – especially when combined with personality measures such as perspective taking and empathic concern. 

We included an explanation on the above in our manuscript on pages 4-5: “The COVID-19 pandemic represents a particularly interesting and unique context to extend our knowledge about adolescents’ giving to various targets, which has so far mainly focused on peers (5,14,17). Specifically, COVID-19 introduces ecologically valid targets that are in need or deserving. These targets of interest include individuals with a poor immune system or COVID-19 symptoms (i.e., targets in need) and medical personnel, such as doctors working in hospitals (i.e., deserving targets). We compared giving to these three targets with giving to an unfamiliar peer (i.e., the default option in most economic games) and a friend (i.e., a target that is more familiar and similar). The reason to examine giving towards targets with verifying degrees of familiarity, similarity, and need and deservedness within individuals was to get more insight in the mechanisms and motivations underlying giving decisions.”

Furthermore, we emphasize our core contributions and main strengths in the discussion on page 26-27: “This study had several strengths. First, we used intensive daily diary assessments during the pandemic. As such, measurements within individuals were based upon multiple assessments which increases robustness and reliability. Second, to draw reliable conclusions about changes in behavior, it is important that measurements are obtained pre- and peri-pandemic, as was the case in our longitudinal project. The within-subject-, longitudinal design enhanced statistical power. Third, this study obtained data in the first weeks of the pandemic, and is therefore unique in capturing the very early experiences of being in lockdown. Fourth, the novel experimental Dictator Games allowed us to disentangle familiarity-, need-, and deservedness-effects on giving in an ecological valid context (i.e., the COVID-19 pandemic). Fifth, this study focused on pandemic effects on adolescents, a relatively understudied age period that is formative with regard to socio-emotional development.”

Reviewer 2

Reviewer #2: Overview

Overview

The study makes multiple, tentatively related observations using a unique survey of adolescents. The authors present one set of findings comparing pre-pandemic and pandemic data: mood improves (vigor increases, tension decreases), empathetic concern decreases, and perspective taking increases. Drawing on data exclusively collected during the pandemic the authors furthermore observe that adolescents state that they would give the same amount of money to healthcare workers as to friends. The authors argue that these results are intimately tied to reduced social contacts during the pandemic. While I think that studying the impact of the pandemic on adolescents is important, I have some concerns which I detail below.

Major Concerns:

1) Connection between motivation and results

a. The motivation is focuses heavily on social contacts which is one important thing that changed during lockdown. But a lot of other important things changed as well, which may have/did affect mood, worries about the future, worries about the health of family members etc. The described link between social contacts and the behaviors investigated is too tentative for me. The authors should give more room to other factors that could also play a role in the introduction. This would also allow them to describe more nuanced expectations (see below).

Answer: We now included these suggestions in the introduction and discussion, see p.5 “Besides social distancing, the pandemic has resulted in many other changes for adolescents, including possible worry about health of family members, fear of death, worry about financial consequences, and worry about one’s future (9,12). Therefore, social changes in the pandemic should be interpreted in the context of multiple system disruptions (13).”

2) Key findings and main contribution

a. The authors present a collection of interesting results. However, it remains unknown to the reader what the authors see as their main contribution. I am aware that the study is exploratory, but it feels like there are too many different results which do not necessarily fit together well. There are two potential solutions: i) a reduction of the number of highlighted results, ii) a more thorough description of how the results are related. I would recommend the authors to focus on a subset of the results they currently present. From my point of view, the before and after comparison on mood, empathetic concern, and perspective taking has the most potential, since the authors have the same data before and during the pandemic. An alternative may be to focus on the giving aspect while highlighting the other results as a support for your findings on giving. In any case, the main contribution of the authors has to be more clearly stated in the introduction.

Answer: We thank the reviewer for the suggestion to streamline the manuscript and highlight main contributions. We made changes to more clearly highlight the two main contributions/aspects of the current study: 1) comparing pre- to peri-pandemic measures of mood, empathy, and prosocial behavior to investigate changes as a result of the COVID-19 pandemic and 2) investigating peri-pandemic effects of target familiarity, need, and deservedness on giving behavior by utilizing novel hypothetical dictator games with ecologically valid targets; and to examine moderating effects of empathy, contributions to society, and opportunities for prosocial actions (i.e., variables that changed as a result of the pandemic). 

This decision was also based on reviewer 1’s recommendations to include all acquired data of which we had pre- and peri-pandemic measures into the manuscript. We hope the rationale is now described more clearly with the stronger focus on the two main goals of the study. 

b. The abstract can lead to misinterpretation. Which results are based on before and after data? How are the different results connected? The reader is a bit lost with all the bits of information. For example, the mood effects are mentioned in passing. Then the authors go on at length describing levels of giving, whereas the reader was expecting the authors to describe changes in giving because of the pandemic. I feel the authors should more clearly highlight the cross-sectional nature of their investigation for the giving results, rather than first implying focusing on changes, but then discussing giving based on levels. In my view, it would help to state clearly which data are available for a before and after comparison and which data were only collected during the pandemic. At the same time, given the different results, the abstract is quite long. If the authors shorten the highlighted set of results it will probably improve the flow of the abstract, too.

Answer: We thank the reviewer for the suggestions and now describe more clearly which results are informed by peri-pandemic measurements only (i.e., the giving results) and which results are informed by both pre- and peri-pandemic measurements. We adjusted the abstract and introduction based on this suggestion. 

3) Giving results

a. The authors do surveys with hypothetical choices. They should make this clear in the abstract and introduction.

Answer: We now more explicitly mentioned this in the abstract and introduction. 

b. The authors should discuss the merits of using a dictator game with multiple recipients and the potential impacts on the results compared to a situation where one game is played with each recipient.

Answer: We used single shot dictator games in which participants divided 10 coins between themselves and another target. As such, they played 5 trials of the game (i.e., one with each target), which were displayed on separate screens and presented in randomized order. We chose this within-subject design over a between-subjects design where each participant encountered only one of the five targets to increase statistical power.

Although most prior studies have focused on anonymous giving, we decided to focus on giving to various targets for several reasons. First of all, giving to completely anonymous others is less prevalent in adolescents’ lives than giving to individuals with whom they have relationships. Second, giving towards targets with various degrees of familiarity and need within the same individuals may give insight into the possibly different mechanisms and motivations underlying these types of giving, especially when investigated in interplay with personality factors such as perspective taking and empathic concern. 

We now emphasized the advantages of our design more clearly in our manuscript, see page 4, “The COVID-19 pandemic represents a particularly interesting and unique context to extend our knowledge about adolescents’ giving to various targets, which has so far mainly focused on peers (5,14,17). Specifically, COVID-19 introduces ecologically valid targets that are in need or deserving. These targets of interest include individuals with a poor immune system or COVID-19 symptoms (i.e., targets in need) and medical personnel, such as doctors working in hospitals (i.e., deserving targets). We compared giving to these three targets with giving to an unfamiliar peer (i.e., the default option in most economic games) and a friend (i.e., a target that is more familiar and similar). The reason to examine giving towards targets with verifying degrees of familiarity, similarity, and need and deservedness within individuals was to get more insight in the mechanisms and motivations underlying giving decisions.”

c. It would be good to have more details on why correlations in giving change during the pandemic, but the levels of giving does not change. How can we reconcile these findings?

Answer: There are indeed some differences in correlations between targets when comparing day 1 and day 15. The most pronounced difference has to do with the correlation between giving to an unfamiliar peer and an individual with a poor immune system. This correlation is .06 (n.s.) on day 1 (N = 53), but .34* on day 15 (N = 36). To get more insight into this difference, we calculated the correlation on day 1 only for the 36 participants who participated on day 15. The correlation was now .35*, suggesting that the change in this correlation may be due to the difference in sample size/the individuals included in the analyses. 

We also created scatterplots (see reply letter attachment) for the association between giving to an unfamiliar peer and individual with a poor immune system on day 1 (N = 53 and N = 36) and on day 15 (N = 36). As can be seen below, the sample of 53 participants shows more variation: there are more adolescents who give nothing to the unfamiliar peer, while donations to the individual with a poor immune system are slightly higher. The repeated measures ANOVAs in which we compared giving levels on day 1 and 15, however, are based on the sample of 36 participants. This sample is more uniform and shows higher giving to an unfamiliar peer, but lower giving to an individual with a poor immune system compared to the N=53 sample on day 1. 

This can reconcile the findings that correlations in giving change during the pandemic, whereas we did not find changes in levels of giving. We interpreted the results of participants who have giving scores on both time points, notwithstanding that there should also be transparency about possible self-selection bias which might contribute to the variation in the findings. 

We describe the limitations with regard to self-selection bias in the discussion on page 27: “Secondly, we invited individuals who participated in a cohort-sequential longitudinal study to participate in the daily diary study during the pandemic. While this allowed us to compare experiences and social behavior prior to and during the COVID-19 pandemic, it could also have introduced self-selection bias and it limited our potential sample size.”. 

However, there are several reasons why this study makes a valuable contribution to the literature. First, we used intensive daily diary assessments, therefore measurements within individuals were based on multiple assessments which increases robustness and reliability. Second, to draw reliable conclusions about changes in behavior, it is important that measurements are obtained prior to and during the pandemic, as is the case in our longitudinal project. Within-subject-, longitudinal designs enhance statistical power, which in our view outweighs the disadvantages of the smaller sample size. Third, this study obtained data in the first weeks of the pandemic, and therefore is unique in capturing the very early experiences of being in lockdown. All strengths of this study are now more clearly described on page 26 of the manuscript. 

d. The possible role of experimenter demand for the hypothetical giving decisions should be discussed more thoroughly. What did the authors do to avoid experimenter demand effects and how would they expect it to affect the results? I do not think that the nature of the response to this point should affect chances for publication, but I do think it should be discussed in the study.

Answer: Based on your feedback as well as the feedback from reviewer 1 (comment 4), we tested whether the target effects in the Dictator Game still held when controlling for social desirability. We did this by adding the social desirability response set (SDRS-5) variable as a covariate to the repeated measures ANOVA within target (5 levels) as a within-subject variable. There was no main effect of social desirability on giving (p = .371), nor was there an interaction between social desirability and target (p = .223). In addition, calculating bivariate correlations between social desirability and giving to each of the five targets revealed no significant associations. 

These findings strengthen our interpretation that the results obtained in this study are not driven by experimenter demand effects. Furthermore, our instructions were carefully designed to prevent experimenter demand effects or norm communication, as participants were explained that there were no right or wrong answers and they could decide upon the division of coins by themselves. Typically, norm driven behavior would lead to equity divisions in zero sum games (Fehr & Fischbacher, 2003; Fehr & Schmidt, 1999; Meuwese, Crone, de Rooij, & Güroglu, 2015), therefore, the larger donations to COVID-19 targets can be interpreted as prosocial behavior. Finally, we displayed each trial (i.e., the donation to each target) on separate screens, which together with the randomized order minimized adjustments based on targets yet to come. The details regarding the instructions are now more explicitly mentioned in the methods (pages 16-17, e.g. “To minimize experimenter demand effects, it was emphasized that there were no right or wrong answers, and each trial (i.e., the donation to each target) was shown on a separate screen.”). 

4) Pre- and post comparison

a. As far as I understand the authors use the observations from the last survey before the pandemic as a benchmark, rather than using all surveys before. The latter would allow the authors to show trends in the variables and then to see whether there is a break in the trend at the time of the pandemic. This would be a more convincing way of showing that there is a substantial change. Otherwise, it could be that there is a general trend in these variables which was not different in the pandemic when compared to before. So what I would like to see would be a figure showing all the raw averages, e.g., for empathetic concern, for each survey before the pandemic and then the average during the pandemic.

Answer: We agree with the reviewer that including all the surveys obtained before the pandemic would give the best indication of change in the variables of interest. This is indeed what we did, both in our analyses and figures. In the schedule below (see reply letter attachment), we show for each of the variables at what point in time they were obtained. In our analyses, we averaged over measurements obtained at T1.5 (i.e., over a maximum of 5 questionnaires). We also averaged over week 1, 2, and 3 with regard to measurements that were obtained during the pandemic on a daily basis (i.e., over a maximum of 5 questionnaires per week). Finally, we averaged over the measures that were obtained 4 times during the pandemic (i.e., perspective taking, empathic concern, general contributions to society; GCS). As such, the following distinct time points were used in analyses: T1, T1.5, T2, Week 1, Week 2 and Week 3 (for daily measures), pandemic average (for empathic concern, perspective taking, and GCS); Day 1 and Day 15 (for giving and SVO). Therefore, all figures and analyses already include all the pre-pandemic measurements that were obtained, not just the last survey before the pandemic. 

5) Exploratory nature of the study

a. I think it is commendable that the authors mention that their study is exploratory in the introduction. I would go even further and clearly highlight this in the abstract. I think it is an interesting exploratory study and can be labelled as such. This will also allow the authors to present a more nuanced perspective on possible influences of the pandemic on adolescents. 

Answer: We now mention the exploratory nature of the study in the abstract and introduction. 

b. Hypotheses/expectations: The expected relationships the authors describe on page 6 seem slightly contradictory and need more justification. The authors first formulate very precise expectations with respect to the before and after lockdown comparisons, but then do not have expectations for the results from the diary study during the lockdown “due to the historical uniqueness of the COVID-19 pandemic”. In my view, the authors could either cut this part entirely or generate a new section where they discuss different plausible expectations and corresponding reasons more exhaustively. I would be happy with either change, I think for the reader the latter might be a bit more attractive, but I am not sure it is worth the additional time by the authors.

Answer: We thank the reviewer this suggestion. We have now cut this part entirely, also because we highlight the exploratory nature of the study throughout the manuscript and especially the introduction. 

6) Attrition: The authors should be more openly stating potential issues with attrition. Why are there many more females taking part? How could this affect the results? How is dropping out of the survey related to pre-pandemic levels of mood and empathetic concern? I do not think the results of this examination should affect the chances for publication, but I do think it is important to discuss these issues openly.

Answer: There are several factors that could explain the attrition in this study. First, the pandemic and lockdown resulted in many changes in the lives of adolescents and their parents, including home schooling, new family routines and online communication. The current study was not previously announced and required participants to sign up within the period of one-week time. This many have caused larger attrition than is typical in longitudinal studies. 

Second, it is noteworthy that attrition was larger for boys than for girls. There is currently no explanation for this difference but it is worth noting that other pandemic studies have also shown more female than male participations (Branas-Garza et al., 2020; Fried, Papanikolaou, & Epskamp, 2020; Kwong et al., 2020; Zettler et al., 2020). 

Based on the reviewer’s suggestion we tested whether participants who dropped out differed from those wo did not in terms of mood and empathic concern. ANOVAs with participation in the COVID-19 daily diary study as a predictor (i.e., yes/no) revealed that there were no differences between the groups with regard to perspective taking, age, emotional support, dire prosociality, vigor, and tension on the first measurement (i.e., T1 in 2018 for most variables; T1.5 in 2018-2019 for vigor/tension). There was, however, a significant difference between the groups in empathic concern, F(1, 131) = 4.53, p = .035, such that participants who partook in the daily diary study scored higher (M = 2.68, SD = .59) than those who did not (M = 2.44, SD = .64). The same was observed for altruistic prosociality, F(1, 131) = 4.12, p = .044, such that participants who participated in the daily diary study scored higher (M = 4.28, SD = .59) than those who did not (M = 4.03, SD = .77). Note that these are the exact p-values, uncorrected for multiple comparisons. This suggests that the participants who participated in the daily diary study were slightly more empathic towards friends and slightly more altruistic prior to the pandemic, but did not show differences on the other variables. 

We now discuss this in the manuscript, see page 31, “Secondly, we invited individuals who participated in a cohort-sequential longitudinal study to participate in the daily diary study during the pandemic. While this allowed us to compare experiences and social behavior prior to and during the COVID-19 pandemic, it could also have introduced self-selection bias (e.g. inclusion of empathic and altruistic individuals) and it limited our potential sample size. The attrition was larger in males, a trend that is also observed in other studies (33,34,36,38). Future studies should employ large, diverse samples to test the robustness of these findings and to test for individual differences.”

7) Range of emotions/mood variables and corresponding measurements a. Why do the authors focus on tension and vigor only? Why are there no other mood variables included? The authors should justify the focus on these two.

Answer: We focused on tension and vigor for the following reasons. First of all, with regard to the three-week daily diary study, we aimed to minimize the time burden on participants as much as possible. Due to these time constraints we decided to focus on vigor, as an indication of adolescent resiliency. Second, we focused on tension (also sometimes called tension-anxiety) as several studies have suggested that the pandemic (i.e., an uncertain and frightening time), as well as social isolation, can increase feelings of tension and anxiety (Orben, Tomova, & Blakemore, 2020). 

We have made our justification for choosing these two mood variables more clearly in the manuscript, see pages 14-15, “Although the original questionnaire included more emotions than vigor and tension, we decided to focus on these subscales as we aimed to focus on a positive emotion (vigor) to get an indication of adolescent resiliency, and a negative emotion (tension) as prior studies have suggested increases in adolescent tension and anxiety as a result of the pandemic (2).”

8) Standard errors

a. The authors should give details on how they compute standard errors. Do they take into account that they have correlations within individuals in the error terms over time? In case the authors do not account for this they should do so for the revision and present all results with corresponding standard errors (often referred to as cluster robust standard errors).

Answer: This is a valid concern but for the current exploratory study we made use of repeated measures ANOVAs in SPSS using ‘General Linear Model � Repeated Measures’ rather than mixed linear models. This analysis accounts for the within-subjects nature (i.e., dependency/relatedness of data within individuals) of the data, but does not calculate cluster robust standard errors. We agree that with larger sample sizes mixed linear models are a better solution, particularly for disentangling general developmental patterns from individual differences in growth trajectories, and we recommend this on page 27: “Sixth, the current study accounted for dependence of observations by using Repeated Measures ANOVAs. Future studies with larger sample sizes should employ mixed linear models to disentangle general developmental patterns from individual differences in growth trajectories.”

Minor concerns:

1) The authors should discuss their results on emotional support given that the measure includes questions strongly related to in-person meetings with friends. I think this is no problem, but I would like to see more on this.

Answer: We apologize for this confusion. The question that we used in Dutch actually was ‘Last day, I did my best to carve out time for friends instead of how we translated it to English in the first version of our manuscript: to spend time with friends. We now adjusted this, such that the question pertains to carving out time for friends, which does not necessitate in-person meetings. 

2) I would like to see more explanation on Table 2 and its interpretation. At the moment it is very cursory. Some of the results seem at odds with the texts where the authors state that the results underscore the robustness of the giving pattern, but when one looks at the table the correlations seem not very high given the short amount of time between the surveys and when compared to correlations across time in other studies on prosociality.

Answer: Giving to the various targets on day 1 shows correlations in the range of .19** - .57** with giving to the same target on day 15, with the exception of the friend (n.s. correlation). To our knowledge, most studies examining within-subject correlations of Dictator Game giving over time have focused mostly on giving to unknown, anonymous others (Baumert, Schlösser & Schmitt, 2014; Haesevoets, Reinders Folmer & Van Hiel, 2015). These studies, in which multiple dictator games were administered in one sitting, revealed correlation estimates in the range of .70 - .91. The relatively lower correlations in this study could be related to time-related variability in perceptions of the target. For example, adolescents could have received new information about certain targets, or could have had new experiences with certain targets. This might also explain why there was no significant associations for the friend over time: it is possible that adolescents thought of a different friend across time or that their relationship changed over the course of three weeks. 

We adjusted the text in our manuscript on page 26: “The novel Dictator Games provided a reliable measure of giving as was shown by significant correlations across time (i.e., r’s .19-.57), although these correlations were lower than those reported in studies where Dictator Games to anonymous others were repeatedly administered in the same sitting (i.e., r’s .70-.91). The only target for which giving was not significantly correlated over time was the friend, which was also the target that showed age-related differences at the final time point. It is also possible that adolescents thought of a different friend across time points or that their relationship changed over the course of three weeks. The correlations over time for the other targets may also have been influenced by time-related variability in perceptions of the targets, especially as these measures were taken in the first weeks of the pandemic.”

3) It would be good to have the distribution of key variables shown in histograms in the supplementary material.

Answer: We included histograms of the key variables in S4 (Supplement 4). 

4) “Even though it is possible to discern general patterns of adolescents’ giving to different targets, prior research has shown that not all adolescents are prone to give to others.” This sentence seems a bit contradictory, I would rewrite it.

Answer: We changed this sentence into: “Despite the general trend that adolescents adjust their levels of giving to the familiarity of a target, there are marked individual differences in how much adolescents give.” 

5) NA in dictator giving: I would be interested in seeing the correlation of contribution to society with NA in the giving game to understand whether NA basically means less prosocial.

Answer: We assume that NA in dictator giving stands for giving nothing to the other (i.e., not applicable). This happened only in trials where the target was an unfamiliar other. To test the reviewer’s suggestion, we recoded DG giving on day 1 to an unfamiliar other into no giving (0) or other (1). An ANOVA revealed no differences between these groups with regard to their general contributions to society, p = .660. 

6) There are a number of other studies on prosociality and COVID-19 focusing on adults that should be referred to in the study:

- Branas-Garza et al. “Exposure to the Covid-19 pandemic and generosity in southern Spain”

- Cappelen et al. “Solidarity and Fairness in Times of Crisis”

- Zettler et al. “Individual differences in accepting personal restrictions to fight the COVID-19 pandemic: Results from a Danish adult sample.

- Campos-Mercade et al. “Prosociality predicts health behaviors during the COVID-19 pandemic”

- Everett et al. “The effectiveness of moral messages on public health behavioral intentions during the COVID-19 pandemic.”

- Bilancini et al “The effect of norm-based messages on reading and understanding COVID-19 pandemic response governmental rules”

Answer: We thank the reviewer for pointing us towards these studies on the links between prosociality and COVID-19 in adults. We incorporated all of these studies into our manuscript:

Banas-Garza et al. & Cappelen et al.: “In contrast to opportunities for prosocial actions, which declined during the pandemic, we found no pre- to peri-pandemic changes in general contributions to society, social value orientation and altruistic and dire prosociality. Our findings suggest no changes in general prosocial tendencies towards unfamiliar others or society as a whole, only a decline in concrete daily prosocial actions towards familiar others (i.e., friends). Interestingly, recent studies in adults suggest that during the pandemic there is a shift towards pandemic-specific prosocial behaviors. For example, these studies have demonstrated that the crisis makes individuals more likely to prioritize society’s problems over their own, but less willing to give to non-COVID-19 related causes (36,37). This interpretation also fits with our finding that Dictator Game giving is higher for COVID-19 related targets. Future studies should aim to replicate this finding in both adults and adolescents and examine whether this shift persists further into and after the pandemic.”

Zettler et al., Campos-Mercade et al., Everett et al., and Bilancini et al.: “Another interesting direction for future research is to link these types of giving with COVID-19 related health behaviors and engagement with government regulations. Recent studies in adults have namely demonstrated various links of pandemic health behaviors with several forms of prosociality (38–41).”

---

## [Decision Letter · Decision Letter 1]

2 Sep 2020

PONE-D-20-19516R1

A daily diary study on adolescents’ mood, empathy, and prosocial behavior during the COVID-19 pandemic

PLOS ONE

Dear Dr. van de Groep,

Thank you for submitting your manuscript to PLOS ONE. After careful consideration, we feel that it has merit but does not fully meet PLOS ONE’s publication criteria as it currently stands. Therefore, we invite you to submit a revised version of the manuscript that addresses the points raised during the review process.

We look forward to receiving your revised manuscript.

Kind regards,

Valerio Capraro

Academic Editor

PLOS ONE

Additional Editor Comments (if provided):

The reviewers still have some minor suggestions before publication. Please address their comments at your earliest convenience. I am looking forward for the final version.

Reviewers' comments:

Reviewer's Responses to Questions

**Comments to the Author**

1. If the authors have adequately addressed your comments raised in a previous round of review and you feel that this manuscript is now acceptable for publication, you may indicate that here to bypass the “Comments to the Author” section, enter your conflict of interest statement in the “Confidential to Editor” section, and submit your "Accept" recommendation.

Reviewer #1: (No Response)

Reviewer #2: (No Response)

2. Is the manuscript technically sound, and do the data support the conclusions?

Reviewer #1: Yes

Reviewer #2: Partly

3. Has the statistical analysis been performed appropriately and rigorously? 

Reviewer #1: Yes

Reviewer #2: N/A

4. Have the authors made all data underlying the findings in their manuscript fully available?

Reviewer #1: Yes

Reviewer #2: Yes

5. Is the manuscript presented in an intelligible fashion and written in standard English?

Reviewer #1: Yes

Reviewer #2: Yes

6. Review Comments to the Author

Reviewer #1: I think the paper improved substantially since the last submission. The authors addressed most of my concerns. I have five suggestion for four changes:

1. “Aim of the study”

In the paper, the authors often write that the “aim of the study was to” i) investigate the effect of the COVID-19 pandemic on adolescents’ mood etc. and ii) investigate the effect of familiarity, need and deservingness on giving behavior. I think this formulation makes most readers belief that these two questions were the initial research questions. However, the study was designed with different research questions in mind (see pre-analysis plan). While the authors mention this in the paper, I think they should avoid saying that “the aim of the study was…” to avoid misunderstandings. (I would prefer, for example, the formulation “this study investigates…”)

2. Correction for multiple hypothesis testing

The authors address the concern of multiple hypothesis testing in a separate paragraph. They write that “most confidence should be placed in the results reported with p‐ values < .006 and <

.001, respectively.” The results section does not always follow this recommendation. For example, on p.18 they write ..“comparisons showed that adolescents displayed significantly higher levels of perspective taking during the COVID-19 pandemic (M = 2.74, SD = .63), compared to […] T2 (M = 2.59, SD = .59, p = .040).” Note that this difference is not significant after acounting for multiple hypothesis testing. I think this should be improved. (The authors could, for example, move the discussion of multiple hypothesis testing at the end of the result sections “research aim 1…” and “research aim 2…”, saying that most results are also significant with p‐ values <.006 and <.001, respectively.)

3. Small sample size

I am still worried about the small sample size of in between 36 and 53. While the authors can not solve this issue, it might be worth mentioning the sample size in the abstract. This would allow readers that only read the abstract to take the sample size into account when judging the paper.

4. Section “Peri-pandemic Individual Differences in Giving”

I don’t think the section on “Peri-pandemic Individual Differences in Giving: Influence of …” is very insightful, in particular due to the small sample size. I personally think the paper would improve by deleting this section (or, moving it to the Appendix).

5. I think the writing of the paper could be improved

Reviewer #2: Overview

The study makes multiple, tentatively related observations using a unique survey of adolescents. The authors present one set of findings comparing pre-pandemic and pandemic data: mood improves (vigor increases, tension decreases), empathetic concern decreases, and perspective taking increases. Drawing on data exclusively collected during the pandemic the authors furthermore observe that adolescents state that they would give the same amount of money to healthcare workers as to friends. The authors argue that these results are intimately tied to reduced social contacts during the pandemic.

The authors put great care in the revision, they reacted to my prior comments and rewrote the paper substantially. I do have some remaining and, in my view, relevant concerns detailed below.

1. Selection of individuals into answering different waves of the survey:

a. I would recommend adding the response of the authors on systematic attrition in response to my previous comments in the manuscript (second paragraph of the answer of the authors to the first report, point 6).

b. I would recommend showing the main figures capturing the results using two samples: Sample 1 includes all individuals are retained in all surveys (no attrition across points in time, only include people that participated in all surveys). Sample 2 includes every observation independent of whether they were answering all surveys.

2. Standard errors:

a. The authors should state explicitly what assumptions they make to compute the standard errors and corresponding arguments for why these assumptions hold in their data. As far as I understand it, the current analysis assumes that within individual correlations in error terms have the same correlation structure across individuals (that is, errors are independent across individuals, but have a similar/the same correlation within individuals). This may not be the case in the data. In any case, the authors should argue why the assumptions they use to compute standard errors / t-values are reasonable. I also think that the authors should relax the assumptions if they are very restrictive and show the corresponding results (e.g., if the above statement on the error structure applies, what happens if the authors assume that the within-individual dependency of errors are not identical across individuals?).

3. Number of emotions:

a. The authors now mention in the text that they only focused on two emotions in the follow-up questionnaires: “Although the original questionnaire included more emotions than vigor and tension, we decided to focus on these subscales as we aimed to focus on a positive emotion (vigor) to get an indication of adolescent resiliency, and a negative emotion (tension) as prior studies have suggested increases in adolescent tension and anxiety as a result of the pandemic (2).” This sentence is not clear to me. It sounds like the authors had more questions in the questionnaire they ended up administering, but I had the impression the authors only included questions on vigor and tension in the questionnaire in the follow-ups. Could the authors clarify which one of these applied? If only vigor and tension were included in the follow-up questionnaires this should be more explicitly stated. If more emotions were collected, this should be added to the statement.

Minor points: The last sentence of the abstract could be sharpened. It seems a bit broad at the moment.

7. PLOS authors have the option to publish the peer review history of their article (what does this mean?). If published, this will include your full peer review and any attached files.

Reviewer #1: No

Reviewer #2: No

---

## [Author Response · Author response to Decision Letter 1]

23 Sep 2020

Editor:

The reviewers still have some minor suggestions before publication. Please address their comments at your earliest convenience. I am looking forward for the final version.

Answer: We thank the editor for the opportunity to revise our article ‘A daily diary study on adolescents’ mood, empathy, and prosocial behavior during the COVID-19 pandemic’. Please find below our replies to the helpful comments we received.

Reviewers' comments:

Reviewer #1:

 I think the paper improved substantially since the last submission. The authors addressed most of my concerns. I have five suggestion for four changes:

1. “Aim of the study”

In the paper, the authors often write that the “aim of the study was to” i) investigate the effect of the COVID-19 pandemic on adolescents’ mood etc. and ii) investigate the effect of familiarity, need and deservingness on giving behavior. I think this formulation makes most readers belief that these two questions were the initial research questions. However, the study was designed with different research questions in mind (see pre-analysis plan). While the authors mention this in the paper, I think they should avoid saying that “the aim of the study was…” to avoid misunderstandings. (I would prefer, for example, the formulation “this study investigates…”)

Answer: We completely agree and adjusted our wording throughout the manuscript. 

2. Correction for multiple hypothesis testing

The authors address the concern of multiple hypothesis testing in a separate paragraph. They write that “most confidence should be placed in the results reported with p‐ values < .006 and <.001, respectively.” The results section does not always follow this recommendation. For example, on p.18 they write ..“comparisons showed that adolescents displayed significantly higher levels of perspective taking during the COVID-19 pandemic (M = 2.74, SD = .63), compared to […] T2 (M = 2.59, SD = .59, p = .040).” Note that this difference is not significant after acounting for multiple hypothesis testing. I think this should be improved. (The authors could, for example, move the discussion of multiple hypothesis testing at the end of the result sections “research aim 1…” and “research aim 2…”, saying that most results are also significant with p‐ values <.006 and <.001, respectively.)

Answer: We adhered to this suggestion and now mention the correction for multiple hypothesis testing and its implications at the end of the relevant result sections, see p. 19 and p. 20-21. 

3. Small sample size

I am still worried about the small sample size of in between 36 and 53. While the authors can not solve this issue, it might be worth mentioning the sample size in the abstract. This would allow readers that only read the abstract to take the sample size into account when judging the paper.

Answer: We now mention the sample size (with and without attrition) in the abstract. 

4. Section “Peri-pandemic Individual Differences in Giving”

I don’t think the section on “Peri-pandemic Individual Differences in Giving: Influence of …” is very insightful, in particular due to the small sample size. I personally think the paper would improve by deleting this section (or, moving it to the Appendix).

Answer: We deleted this section and mention in the discussion that future studies should employ larger sample sizes to study individual differences. 

5. I think the writing of the paper could be improved

Answer: We re-read our paper and improved the writing, also based on feedback from a native speaker. 

Reviewer #2:

 Overview

The study makes multiple, tentatively related observations using a unique survey of adolescents. The authors present one set of findings comparing pre-pandemic and pandemic data: mood improves (vigor increases, tension decreases), empathetic concern decreases, and perspective taking increases. Drawing on data exclusively collected during the pandemic the authors furthermore observe that adolescents state that they would give the same amount of money to healthcare workers as to friends. The authors argue that these results are intimately tied to reduced social contacts during the pandemic.

The authors put great care in the revision, they reacted to my prior comments and rewrote the paper substantially. I do have some remaining and, in my view, relevant concerns detailed below.

1. Selection of individuals into answering different waves of the survey:

a. I would recommend adding the response of the authors on systematic attrition in response to my previous comments in the manuscript (second paragraph of the answer of the authors to the first report, point 6).

Answer: We included the response to this comment in the manuscript, see page 26-27: “Second, we invited adolescents enrolled in a cohort-sequential longitudinal study to participate in the daily diary study during the pandemic. While this allowed us to compare experiences and social behavior prior to and during the COVID-19 pandemic, it could have introduced self-selection bias (e.g. inclusion of empathic and altruistic individuals) and it limited our potential sample size. There are several factors that could explain the attrition in this study. First, the pandemic and lockdown resulted in many changes in the lives of adolescents and their parents, including home schooling, new family routines and online communication. The current study was performed ad hoc and required participants to sign up within the period of one week. This may have caused larger attrition than is typical in longitudinal studies. Second, it is noteworthy that attrition was larger for males than for females. There is currently no explanation for this difference, but it is worth noting that other pandemic studies have also shown higher participation of females compared to males [34,35,37,39]. To investigate the extent to which self-selection bias influenced our results, we examined whether adolescents who did not participate in the daily diary study differed from those who did, see S8. These analyses showed that adolescents who participated in the daily diary study showed slightly higher pre-pandemic levels of empathic concern and altruism, but did not differ in terms of mood, perspective-taking, age, and dire prosociality. Future studies should employ large, diverse samples to test the robustness of our findings and to test for individual differences (e.g., in giving)”. 

Details regarding these analyses are provided in supplement 8. 

b. I would recommend showing the main figures capturing the results using two samples: Sample 1 includes all individuals are retained in all surveys (no attrition across points in time, only include people that participated in all surveys). Sample 2 includes every observation independent of whether they were answering all surveys.

Answer: We thank the reviewer for this suggestion and have changed the main figures to include the sample without attrition (i.e., only including individuals who show no attrition over time) and the sample with attrition (i.e., including all available data per time point). 

2. Standard errors:

a. The authors should state explicitly what assumptions they make to compute the standard errors and corresponding arguments for why these assumptions hold in their data. As far as I understand it, the current analysis assumes that within individual correlations in error terms have the same correlation structure across individuals (that is, errors are independent across individuals, but have a similar/the same correlation within individuals). This may not be the case in the data. In any case, the authors should argue why the assumptions they use to compute standard errors / t-values are reasonable. I also think that the authors should relax the assumptions if they are very restrictive and show the corresponding results (e.g., if the above statement on the error structure applies, what happens if the authors assume that the within-individual dependency of errors are not identical across individuals?).

Answer: We thank the reviewer for further explaining this point about standard errors. Based on a literature search we now see that RM ANOVAs are relatively inflexible in modeling within-person correlations and that this can lead to biased and less-precise estimates when correlations are unequal across repeated measures (Muth, Bales, Hinde, Maninger, Mendoza, & Ferrer, 2016). This is especially true for small sample sizes, where standard errors are inherently larger. Based on the reviewers’ suggestions, we now performed analyses using generalized estimating equations (GEE). This analysis is suitable for small samples, better equipped to deal with missing data at certain time points, and allows for various types of correlation matrices. 

For each variable of interest for which we examined time effects, we inspected which correlation structure provided the best goodness of fit as indicated by the lowest QIC value. Specifically, we tested whether each model was best described by 1) an independent correlation matrix (i.e., zero correlation over time), 2) an exchangeable correlation matrix (i.e., constant correlation over time), 3) an autoregressive correlation matrix (i.e., diminishing correlation over time), and 4) an unstructured correlation matrix (i.e., a freely estimated correlation without constraints. The best fitting model per analysis is mentioned in the manuscript (p. 17 – 20) and further described in S7. 

As discussed in Muth et al. (2016): “GEE models are robust to the misspecification of the correlations structure (Zeger & Liang, 1986). Additionally, selecting robust standard errors (Huber/White Sandwich Estimators; as opposed to conventional standard errors) will allow the estimates to be valid even in the event of misspecification of the correlation structure (StataCorp, 2003).” Therefore, we now report robust standard errors throughout the manuscript. Results using GEE models were highly similar to results previously obtained using repeated measures ANOVAs. Nonetheless, as GEE models are more robust and flexible, we decided to report the GEE results in the main text of the manuscript. 

3. Number of emotions:

a. The authors now mention in the text that they only focused on two emotions in the follow-up questionnaires: “Although the original questionnaire included more emotions than vigor and tension, we decided to focus on these subscales as we aimed to focus on a positive emotion (vigor) to get an indication of adolescent resiliency, and a negative emotion (tension) as prior studies have suggested increases in adolescent tension and anxiety as a result of the pandemic (2).” This sentence is not clear to me. It sounds like the authors had more questions in the questionnaire they ended up administering, but I had the impression the authors only included questions on vigor and tension in the questionnaire in the follow-ups. Could the authors clarify which one of these applied? If only vigor and tension were included in the follow-up questionnaires this should be more explicitly stated. If more emotions were collected, this should be added to the statement.

Answer: Indeed, we measured only vigor and tension in the daily diary study, whereas we measured all emotions of the original POMS questionnaire at T1.5. This is now emphasized on page 14: “To measure mean levels and fluctuations in vigor and tension, we used the Profiles of Mood States (POMS; [23]). Although the original questionnaire includes more emotions than vigor and tension, these were not measured during the daily diary study (i.e., at T1.5 we did include all emotions as measured by the POMS). We decided to focus only on vigor and tension subscales in the daily diary study to shorten this study’s length and minimize pressure for participants. We included a positive emotion (vigor) to get an indication of adolescent resiliency, and a negative emotion (tension) as prior studies have suggested increases in adolescent tension and anxiety as a result of the pandemic [2].”

Minor points: The last sentence of the abstract could be sharpened. It seems a bit broad at the moment.

Answer: We changed to last sentences of the abstract to: “This suggests that during the pandemic need and deservedness had a greater influence on adolescent giving than familiarity. Overall, this study suggests detrimental effects of the first weeks of lockdown on adolescents’ empathic concern and opportunities for prosocial actions, which are important predictors of healthy socio-emotional development and resilience. Conversely, adolescents also showed marked resilience and a willingness to benefit others as a result of the lockdown, as evidenced by improved perspective taking and mood, and high sensitivity to need and deservedness in giving to others.”

---

## [Editor Report · Decision Letter 2]

25 Sep 2020

A daily diary study on adolescents’ mood, empathy, and prosocial behavior during the COVID-19 pandemic

PONE-D-20-19516R2

Dear Dr. van de Groep,

We’re pleased to inform you that your manuscript has been judged scientifically suitable for publication and will be formally accepted for publication once it meets all outstanding technical requirements.

Kind regards,

Valerio Capraro

Academic Editor

PLOS ONE
---

## [Editor Report · Acceptance letter]

30 Sep 2020

PONE-D-20-19516R2 

A daily diary study on adolescents’ mood, empathy, and prosocial behavior during the COVID-19 pandemic 

Dear Dr. van de Groep:

I'm pleased to inform you that your manuscript has been deemed suitable for publication in PLOS ONE. Congratulations! Your manuscript is now with our production department. 

Kind regards, 

on behalf of

Dr. Valerio Capraro 

Academic Editor

PLOS ONE